# NAIP–NLRC4-deficient mice are susceptible to shigellosis

Patrick S Mitchell[1†], Justin L Roncaioli[1†], Elizabeth A Turcotte[1], Lisa Goers[2,3,4], Roberto A Chavez[1], Angus Y Lee[5], Cammie F Lesser[2,3,4], Isabella Rauch[6], Russell E Vance[1,5,7,8]*

[1]Division of Immunology & Pathogenesis, Department of Molecular & Cell Biology, University of California, Berkeley, Berkeley, United States; [2]Department of Microbiology, Harvard Medical School, Boston, United States; [3]Broad Institute of Harvard and MIT, Cambridge, United States; [4]Department of Medicine, Division of Infectious Diseases, Massachusetts General Hospital, Boston, United States; [5]Cancer Research Laboratory, University of California, Berkeley, Berkeley, United States; [6]Department of Molecular Microbiology and Immunology, Oregon Health and Science University, Portland, United States; [7]Immunotherapeutics and Vaccine Research Initiative, University of California, Berkeley, Berkeley, United States; [8]Howard Hughes Medical Institute, University of California, Berkeley, Berkeley, United States

*For correspondence:
rvance@berkeley.edu

[†]These authors contributed equally to this work

**Abstract** Bacteria of the genus *Shigella* cause shigellosis, a severe gastrointestinal disease that is a major cause of diarrhea-associated mortality in humans. Mice are highly resistant to *Shigella* and the lack of a tractable physiological model of shigellosis has impeded our understanding of this important human disease. Here, we propose that the differential susceptibility of mice and humans to *Shigella* is due to mouse-specific activation of the NAIP–NLRC4 inflammasome. We find that NAIP–NLRC4-deficient mice are highly susceptible to oral *Shigella* infection and recapitulate the clinical features of human shigellosis. Although inflammasomes are generally thought to promote *Shigella* pathogenesis, we instead demonstrate that intestinal epithelial cell (IEC)-specific NAIP–NLRC4 activity is sufficient to protect mice from shigellosis. In addition to describing a new mouse model of shigellosis, our results suggest that the lack of an inflammasome response in IECs may help explain the susceptibility of humans to shigellosis.

## Introduction

*Shigella* is a genus of Gram-negative enterobacteriaceae that causes ~269 million infections and ~200,000 deaths annually, a quarter of which are of children under the age of five (*Khalil et al., 2018*). Disease symptoms include fever, abdominal cramping, and inflammatory diarrhea characterized by the presence of neutrophils and, in severe cases, blood (*Kotloff et al., 2018*). There is no approved vaccine for *Shigella* and antibiotic resistance continues to rise (*Ranjbar and Farahani, 2019*). *Shigella* pathogenesis is believed to be driven by bacterial invasion, replication, and spread within colonic intestinal epithelial cells (IECs). *Shigella* virulence requires a plasmid-encoded type III secretion system (T3SS) that injects ~30 effectors into host cells (*Schnupf and Sansonetti, 2019*; *Schroeder and Hilbi, 2008*). The virulence plasmid also encodes IcsA, a bacterial surface protein that nucleates host actin at the bacterial pole to propel the pathogen through the host cell cytosol and into adjacent epithelial cells (*Bernardini et al., 1989*; *Goldberg and Theriot, 1995*).

A major impediment to studying *Shigella* is the lack of experimentally tractable *in vivo* models that accurately recapitulate human disease after oral inoculation. Although the infectious dose for

*Shigella* in humans can be as low as 10–100 bacteria (*DuPont et al., 1969*; *DuPont et al., 1989*), mice are resistant to high doses of oral *Shigella* challenge (*Freter, 1956*; *McGuire and Floyd, 1958*). Rabbits, guinea pigs, zebrafish, piglets, and macaques have been used to model human infection (*Islam et al., 2014*; *Jeong et al., 2010*; *Mostowy et al., 2013*; *Ranallo et al., 2014*; *Shim et al., 2007*; *West et al., 2005*; *Yum and Agaisse, 2020*; *Yum et al., 2019*) but the cost and/ or limited tools in these systems impair detailed studies of pathogenesis. Oral streptomycin administration and other treatments facilitate *Shigella* colonization of the mouse intestinal lumen by ablating the natural colonization resistance provided by the microbiome (*Freter, 1956*; *Martino et al., 2005*; *Q S Medeiros et al., 2019*). However, antibiotic-treated mice do not present with key hallmarks of human disease, likely due to the failure of *Shigella* to invade and/or establish a replicative niche within the mouse intestinal epithelium.

Inflammasomes are cytosolic multi-protein complexes that initiate innate immune responses upon pathogen detection or cellular stress (*Lamkanfi and Dixit, 2014*; *Rathinam and Fitzgerald, 2016*). The NAIP–NLRC4 inflammasome is activated when bacterial proteins, such as flagellin or the rod and needle proteins of the T3SS apparatus, are bound by NAIP family members. Importantly, the *Shigella* T3SS inner rod (MxiI) and needle (MxiH) proteins are both potent agonists of human and mouse NAIPs (*Reyes Ruiz et al., 2017*; *Yang et al., 2013*). Activated NAIPs then co-assemble with NLRC4 to recruit and activate the Caspase-1 (CASP1) protease (*Vance, 2015*; *Zhao and Shao, 2015*). CASP1 then cleaves and activates the pro-inflammatory cytokines IL-1β and IL-18 and the pore-forming protein Gasdermin-D (*Kayagaki et al., 2015*; *Shi et al., 2015*), initiating a lytic form of cell death called pyroptosis. We and others recently demonstrated that activation of NAIP–NLRC4 in IECs further mediates the cell-intrinsic expulsion of infected epithelial cells from the intestinal monolayer (*Rauch et al., 2017*; *Sellin et al., 2014*). In the context of *Shigella* infection, it is generally accepted that inflammasome-mediated pyroptosis of infected macrophages promotes pathogenesis by initiating inflammation, and by releasing bacteria from macrophages, allowing the bacteria to invade the basolateral side of intestinal epithelial cells (*Ashida et al., 2014*; *Lamkanfi and Dixit, 2010*; *Schnupf and Sansonetti, 2019*). However, it has not been possible to test the role of inflammasomes in the intestine after oral *Shigella* infection due to the lack of a genetically tractable model. Here, we develop the first oral infection mouse model for *Shigella* infection that recapitulates human disease, and demonstrate a specific host-protective function for inflammasomes in intestinal epithelial cells.

## Results

### *Shigella* appears to suppress the human NAIP–NLRC4 inflammasome

The *Shigella* T3SS effector OspC3 inhibits cytosolic LPS sensing by the human Caspase-4 (CASP4) inflammasome, but is reported not to bind to the mouse ortholog, Caspase-11 (CASP11) (*Kobayashi et al., 2013*). We reasoned that inflammasome inhibition may be a general strategy used by *Shigella* to establish infection, and that such inhibition might occur in a host-specific manner. To test this hypothesis, we compared inflammasome-dependent cell death following *Shigella* infection of mouse C57BL/6J (B6) bone marrow-derived macrophages (BMMs) and human PMA-differentiated THP1 cells. Infection with the wild-type (WT) *Shigella flexneri* strain 2457T but not the avirulent BS103 strain (which lacks the virulence plasmid) resulted in CASP1-dependent cell death in both mouse (*Sandstrom et al., 2019*) and human cells (*Figure 1A,B*). As reported previously, cell death was negligible in *Shigella*-infected mouse *Nlrc4*⁻/⁻ BMMs (*Suzuki et al., 2014*; *Figure 1A*). In contrast, *Shigella* infection induced similar levels of cell death in WT and *NLRC4*⁻/⁻ THP1 cells, indicating that the NAIP–NLRC4 inflammasome is not essential for *Shigella*-induced CASP1 activation in human cells (*Figure 1B*).

To confirm prior reports (*Yang et al., 2013*) that the *Shigella* T3SS MxiH (needle) protein is a potent agonist of the human NAIP–NLRC4 inflammasome, and that THP1 cells express a functional NAIP–NLRC4 inflammasome (*Kortmann et al., 2015*; *Reyes Ruiz et al., 2017*), we produced recombinant LFn-MxiH for cytosolic delivery to cells via the protective antigen (PA) channel, as previously described (*Rauch et al., 2017*; *Rauch et al., 2016*; *von Moltke et al., 2012*). Treatment of THP1 cells with LFn-MxiH+PA ('NeedleTox'), but not PA alone, induced a CASP1- and NLRC4-dependent cell death (*Figure 1C*). We also confirmed that *Shigella* MxiH and MxiI (rod) activate human NAIP–

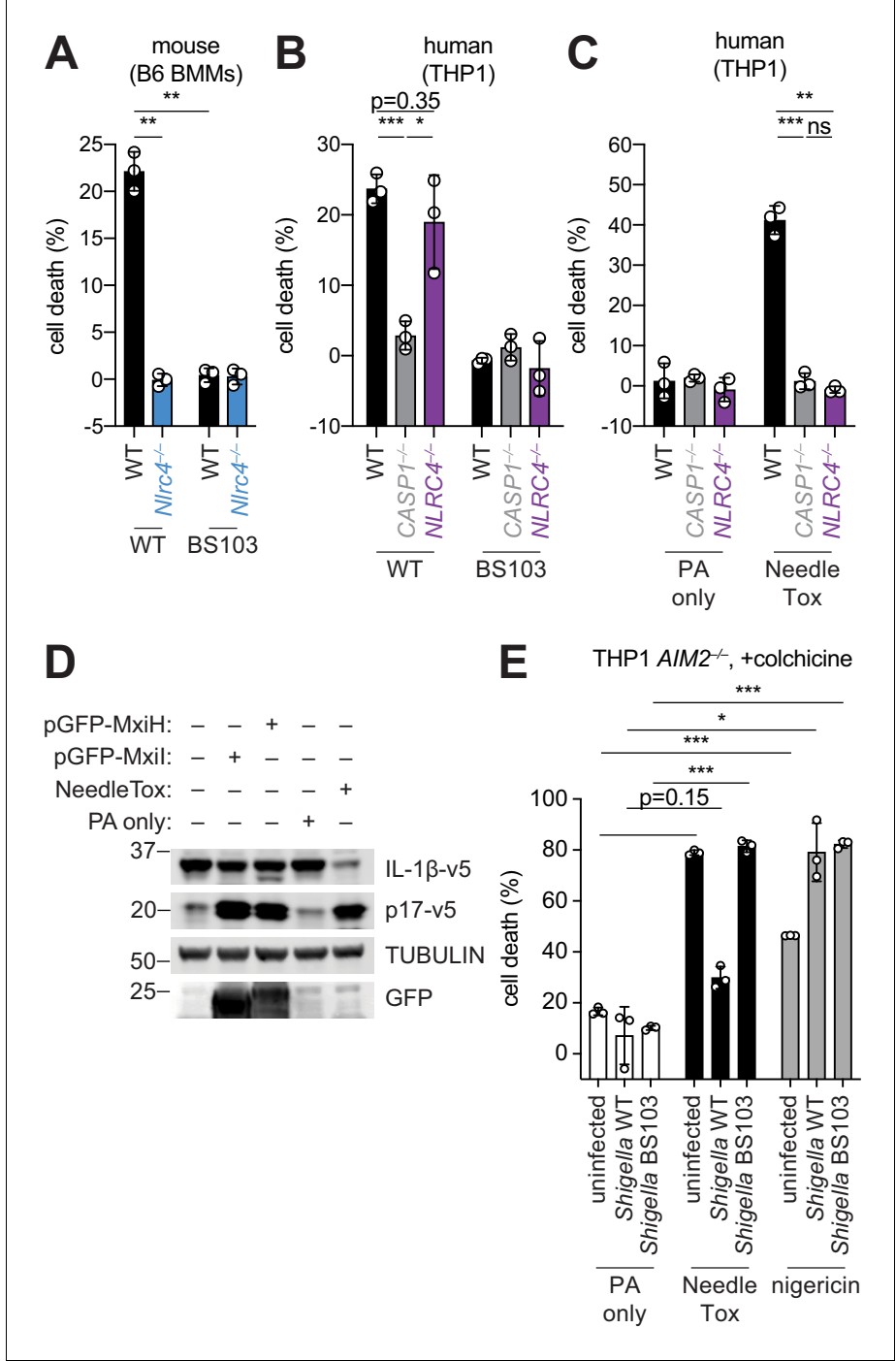

**Figure 1.** *Shigella* infection appears to suppress the NAIP–NLRC4 inflammasome. (**A**) *Shigella* infection (MOI 10) of C57BL/6 WT or *Nlrc4*$^{-/-}$ bone-marrow-derived macrophages (BMMs). Cell death was measured 30 min post-infection (after spinfection, invasion, and washes) by propidium iodide uptake and reported as percent death relative to 100% killing by treatment with Triton X-100. (**B**) Cell death of *Shigella* infected THP1 WT, *CASP1*$^{-/-}$ or *NLRC4*$^{-/-}$ cells as in (A). Cell death was measured 30 min post-infection. (**C**) Cell death of THP1 WT, *CASP1*$^{-/-}$ or *NLRC4*$^{-/-}$ cells treated with 10 µg/mL PA alone or in combination with 10 µg/mL LFn-MxiH ('NeedleTox'). Cell death was measured 4 hr post-challenge. (**D**) Human NAIP–NLRC4 inflammasome reconstitution in 293T cells. Inflammasome activation was measured by CASP1-dependent processing of pro-IL-1β to p17 by co-transfection of an empty vector, pGFP-MxiH or pGFP-MxiI, or by treatment with 10 µg/mL PA alone or in combination with 10 µg/mL LFn-MxiH. (**E**) Colchicine (1 µM)-treated *AIM2*$^{-/-}$ THP1 cells were either left uninfected or infected for 1 hour (after spinfection, invasion, and washes) with WT or BS103 Shigella (MOI 10), and then treated with 10 µg/mL PA

*Figure 1 continued on next page*

*Figure 1 continued*

alone, PA + 1.0 µg/mL LFn-MxiH ('NeedleTox'), or 10 µM nigericin. Cell death was measured by PI staining and is reported as cell death relative to TX-100-treated controls per infection type. Data are representative of at least three independent experiments. Mean ± SD is shown, unpaired t-test with Welch's correction: *p < 0.01, **p < 0.001, ***p < 0.0001, ns = not significant (p > 0.05).

NLRC4 in a reconstituted inflammasome assay (*Reyes Ruiz et al., 2017*; *Tenthorey et al., 2014*; *Figure 1D*). These results confirm that the *Shigella* rod and needle proteins are capable of activating the human NAIP–NLRC4 inflammasome. Moreover, since we observe robust activation of NAIP–NLRC4 in mouse cells (*Figure 1A*), our data indicate that the *Shigella* T3SS needle and rod proteins are delivered to the cytosol during infection. However, the presence of other inflammasomes in THP1 cells likely obscures our ability to determine if *Shigella* activates human NLRC4.

As a strategy to eliminate cell death induced by the AIM2 and PYRIN inflammasomes, we used THP1 *AIM2^{−/−}* cells treated with colchicine, an inhibitor of PYRIN (*Gao et al., 2016*). Interestingly, WT *Shigella* infection did not induce pyroptosis of colchicine-treated *AIM2^{−/−}* THP1 cells (*Figure 1E*). Thus, although human THP1 cells express functional NAIP–NLRC4, and *Shigella* rod and needle proteins can activate NAIP–NLRC4, we nevertheless observe no such activation during *Shigella* infection. We therefore hypothesized that *Shigella* might antagonize the human NAIP–NLRC4 inflammasome. To test this hypothesis, *AIM2^{−/−}* colchicine-treated THP1 cells were either uninfected or infected with the *Shigella* WT or BS103 strains for 1 hr, and then treated with NeedleTox to induce NAIP–NLRC4-dependent cell death. Interestingly, NeedleTox-induced pyroptosis was significantly reduced in cells previously infected with WT but not avirulent BS103 *Shigella* (*Figure 1E*). As a control, treatment with nigericin, an agonist of the NLRP3 inflammasome, induced cell death similarly in cells that were uninfected or infected with either WT or BS103 *Shigella* strains. Thus, WT *Shigella* infection appears to specifically suppress activation of the human NAIP–NLRC4 inflammasome in THP1 cells, leading us to hypothesize that NAIP–NLRC4 activation may be a mouse-specific mechanism of resistance to shigellosis. Future studies will be required to confirm these results with further genetic studies, and to address the mechanism of *Shigella* antagonism of human NAIP–NLRC4, as well as the mechanism of *Shigella* activation of AIM2 and/or PYRIN.

## B6.*Naip*-deficient mice are susceptible to shigellosis

The mouse NAIP–NLRC4 and CASP11 inflammasomes protect the intestinal epithelium from *Salmonella* (*Crowley et al., 2020*; *Rauch et al., 2017*; *Sellin et al., 2014*). Thus, the above experiments led us to hypothesize that the failure of *Shigella* to antagonize mouse inflammasomes might explain the inborne resistance of mice versus humans to *Shigella* infection. A prediction from this hypothesis is that mice lacking inflammasomes might be susceptible to oral *Shigella* challenge. Previous studies have demonstrated that the intestinal microbiota of mice provides intrinsic colonization resistance to diverse enteric pathogens, including *Shigella* (*Ducarmon et al., 2019*; *Freter, 1956*; *Martino et al., 2005*; *Q S Medeiros et al., 2019*). Thus, to overcome this barrier, we pretreated B6 WT mice orally with streptomycin antibiotic. Consistent with prior studies, we found that antibiotic pre-treatment followed by oral infection allows for robust *Shigella* colonization of the intestinal lumen and feces compared to water-only controls (*Figure 2—figure supplement 1*). However, these high lumenal bacterial loads (>10^8 CFU/g feces) did not cause overt disease (*Figure 2* and *Figure 2—figure supplement 2*). To determine if inflammasomes contribute to the resistance of mice to *Shigella*, we orally challenged WT and *Casp1/11^{−/−}* mice with 5 × 10^7 CFU of *Shigella*. CASP1 has been previously reported to drive acute inflammation and *Shigella* clearance during mouse lung infection via the processing and release of IL-1β and IL-18 (*Sansonetti et al., 2000*). Mice that lack CASP1 and/or CASP11 are also more susceptible to oral infection with *Salmonella typhimurium* (*Crowley et al., 2020*). In contrast, B6 WT and *Casp1/11^{−/−}* mice were similarly resistant to *Shigella* infection, showing no signs of intestinal inflammation or disease (*Figure 2—figure supplement 2A–C*). Thus, neither of the primary caspases associated with the canonical or non-canonical inflammasome are essential for resistance to *Shigella* in the mouse intestine. The NAIP–NLRC4 inflammasome can recruit Caspase-8 (CASP8) in the absence of CASP1, an event which leads to non-lytic cell death and delayed IEC expulsion (*Rauch et al., 2017*). We reasoned that this compensatory capacity of CASP8 may account for the resistance of *Casp1/11^{−/−}* mice to *Shigella* infection. Thus, to test directly if the

mouse NAIP–NLRC4 inflammasome mediates resistance to shigellosis, we orally infected streptomycin-pretreated B6 WT and ΔNaip mice with 5 × 10⁷ CFU of *Shigella*. ΔNaip mice (also called *Naip1-6*$^{Δ/Δ}$ mice *Rauch et al., 2016*) harbor a large chromosomal deletion that eliminates expression of all mouse *Naip* genes. Remarkably, *Shigella*-infected ΔNaip but not WT mice exhibited clear signs of disease (*Figure 2*). At 2 days post-challenge, ΔNaip mice had altered stool consistency (*Figure 2A*), cecum shrinkage, and thickening of the cecum and colon tissue (*Figure 2A,B*). Histological analysis of primary sites of infection (cecum, colon) revealed edema, epithelial hyperplasia, epithelial sloughing, and inflammatory infiltrate (predominantly neutrophils and mononuclear cells in the submucosa and mucosa) exclusively in ΔNaip mice (*Figure 2C,D*). In contrast, we did not observe any indicators of inflammation in ΔNaip mice infected with the avirulent BS103 strain (*Figure 2C,D*).

A defining feature of human shigellosis is the presence of neutrophils in patient stools (*Raqib et al., 2000*). The levels of myeloperoxidase (MPO, a neutrophil marker) were low or undetectable in the feces of mice following antibiotic treatment (*Figure 2E*), indicating that microbiota

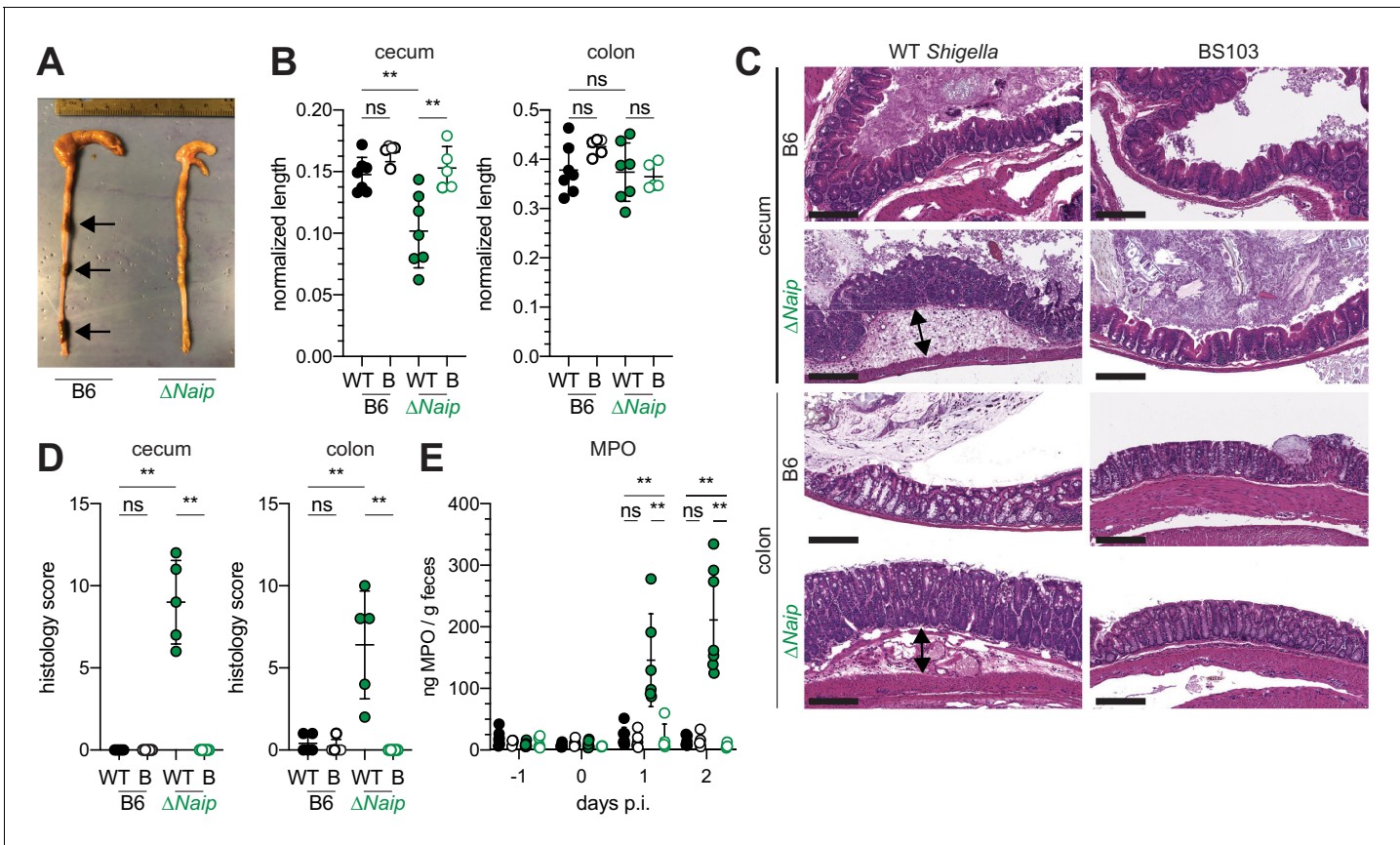

**Figure 2.** *Shigella*-infected B6.ΔNaip mice exhibit intestinal inflammation. (A–E) B6.WT and B6.ΔNaip (green) mice (lacking expression of all *Naip* genes) treated orally with 25 mg streptomycin sulfate were orally challenged the next day with 5x10⁷ CFU of WT or BS103 ('B', non-invasive) *Shigella*. Endpoint harvests were performed at 48 hr post-infection (p.i.). (A) Representative images of the cecum and colon dissected from B6.WT and B6.ΔNaip mice. Note cecum tissue thickening (size reduction), macroscopic edema, and loose stool (absence of arrows). (B) Quantification of cecum and colon lengths. Values were normalized to mouse weight prior to infection; cecum length (cm) / mouse weight (g). WT, wild-type *Shigella* (filled symbols); B, BS103 (open symbols). (C) Representative images of H&E stained cecum and colon tissue from infected mice. Scale bar, 200 µm. (D) Blinded quantification of histology score (cumulative) for tissues in (C). Edema, hyperplasia, inflammatory infiltrate, and epithelial cell death were scored from 0 to 4. The final score is the sum of individual scores from each category. (E) MPO levels measured by ELISA from feces of B6.WT and B6.ΔNaip mice collected -1 through 2 days p.i. (B, D, E) Each symbol represents one mouse. Filled symbols, WT *Shigella*; open symbols, BS103. Data are representative of two independent experiments. Mean ± SD is shown in (B,D,E), Mann-Whitney test, *p < 0.05, **p <0.01, ***p < 0.001, ns = not significant (p > 0.05).

The online version of this article includes the following figure supplement(s) for figure 2:

**Figure supplement 1.** Antibiotic pre-treatment followed by oral route *Shigella* infection permits substantial lumenal colonization.
**Figure supplement 2.** B6.Casp1/11⁻/⁻ mice are resistant to oral *Shigella* challenge.

disruption did not itself promote neutrophilic inflammation. Following *Shigella* infection, however, fecal MPO from Δ*Naip* mice dramatically increased (*Figure 2E*). In contrast, MPO levels remained low in both B6 WT and *Casp1/11*$^{-/-}$ mice (*Figure 2—figure supplement 2D*) or Δ*Naip* mice infected with the avirulent BS103 strain (*Figure 2E*). These results indicate that NAIP–NLRC4-deficient mice experience robust neutrophilic infiltrate consistent with human shigellosis.

## B6.*Nlrc4*$^{-/-}$ mice are susceptible to shigellosis

To confirm that the NAIP–NLRC4 inflammasome confers resistance to *Shigella*, and to control for potential microbiota-associated phenotypes, we next infected streptomycin-pretreated B6.*Nlrc4*$^{+/-}$ and B6.*Nlrc4*$^{-/-}$ littermates, as well as B6 WT mice that had been co-housed for 3 weeks prior to inoculation (*Nlrc4*$^{+/-}$ and B6 WT mice are hereby referred to collectively as *Nlrc4*$^{+}$). Consistent with our prior results in Δ*Naip* mice, we observed thickening of the intestinal mucosa (*Figure 3A*), cecum shrinkage (*Figure 3A,B*), increased fecal MPO levels (*Figure 3C*), and acute weight loss (*Figure 3D*) in *Shigella*-infected B6.*Nlrc4*$^{-/-}$ mice but not B6.*Nlrc4*$^{+}$ littermates or co-housed mice. B6.*Nlrc4*$^{-/-}$ mice also had diarrhea, which was apparent by visual inspection of lumenal contents and quantified by the wet-to-dry ratio of fecal pellets (*Figure 3E*). Thus, B6.*Nlrc4*$^{-/-}$ mice phenocopy the disease susceptibility of B6.Δ*Naip* mice, and strongly suggest that the NAIP–NLRC4 inflammasome mediates the resistance of mice to *Shigella* infection.

To compare the cytokine profiles of B6.*Nlrc4*$^{+}$ and B6.*Nlrc4*$^{-/-}$ mice, we performed ELISAs for pro-inflammatory cytokines IL-1β, IL-18, and the mouse chemokine and neutrophil attractant KC (CXCL1). IL-18 levels were similarly elevated in B6.*Nlrc4*$^{+}$ and B6.*Nlrc4*$^{-/-}$ mice relative to uninfected mice (*Figure 3F*). By contrast, IL-1β and KC levels were significantly elevated in B6.*Nlrc4*$^{-/-}$ relative to B6.*Nlrc4*$^{+}$ and uninfected mice (*Figure 3F*), consistent with the neutrophilic inflammation observed by histology and fecal MPO ELISAs. Since IL-1β is a pro-inflammatory cytokine released primarily from pyroptotic macrophages, we hypothesize that additional inflammasomes are likely active in this cell-type in the absence of NAIP–NLRC4 and may play a role in driving pathogenesis.

Surprisingly, despite the clear differences in disease between *Shigella* infected B6.*Nlrc4*$^{+}$ and B6.*Nlrc4*$^{-/-}$ mice, we found no significant difference in the bacterial burdens of whole cecum or colon tissue (*Figure 3G*). To more directly measure the intracellular colonization of IECs, the primary replicative niche for *Shigella*, we enriched IECs from the ceca and colons of infected B6.*Nlrc4*$^{+}$ and B6.*Nlrc4*$^{-/-}$ mice (see Methods). We found an ~20-fold difference in colonization in enriched IECs between B6.*Nlrc4*$^{+}$ and B6.*Nlrc4*$^{-/-}$ mice (*Figure 3H*), indicating that disease in our model correlates with invasion of and replication in IECs. Importantly, we observed no CFU differences in feces at the time of harvest (*Figure 3—figure supplement 1A*), excluding the possibility that differences in IEC CFU were merely the result of differences in lumenal *Shigella* density. These data suggest that *Shigella* colonizes the intestinal tissue regardless of inflammasome activity, but can only specifically invade the epithelium and provoke disease in NAIP–NLRC4-deficient mice.

## *Shigella* causes bloody diarrheal disease in 129.*Nlrc4*$^{-/-}$ mice

To determine whether the role of NAIP–NLRC4 in mediating protection against *Shigella* is robust across diverse mouse strains, we generated *Nlrc4*$^{-/-}$ mice on the 129S1 genetic background. 129.*Nlrc4*$^{-/-}$ mice have a 10 bp deletion in exon 5 of the *Nlrc4* coding sequence, resulting in loss of NLRC4 function (*Figure 4—figure supplement 1*). Importantly, 129S1 mice are naturally deficient in CASP11, which responds to cytosolic LPS, and thus 129.*Nlrc4*$^{-/-}$ mice lack functional NAIP–NLRC4 and CASP11 signaling. Similar to the B6 genetic background, antibiotic-pretreated 129.*Nlrc4*$^{-/-}$ but not 129.*Nlrc4*$^{+/-}$ littermates challenged with WT (or BS103) *Shigella* exhibited severe signs of shigellosis, including pronounced edema, epithelial cell hyperplasia, and disruption of the columnar epithelium of infected tissues (*Figure 4A,B*). 129.*Nlrc4*$^{-/-}$ mice also exhibited dramatic cecum shrinkage and diarrhea (*Figure 4C,D,K*), lost between 8% and 18% of their starting weight within 2 days of infection (*Figure 4E*), and exhibited a massive increase in fecal MPO following infection (*Figure 4F*). We found no significant difference in the bacterial colonization of the whole cecum and colon tissue between 129.*Nlrc4*$^{+/-}$ and 129.*Nlrc4*$^{-/-}$ mice (*Figure 4G*). However, IECs enriched from infected 129.*Nlrc4*$^{-/-}$ mice again exhibited ~20-fold higher bacterial burdens than IECs enriched from 129.*Nlrc4*$^{+/-}$ mice (*Figure 4H*), despite similar levels of lumenal colonization (*Figure 3—figure supplement 1B*). A hallmark of severe human shigellosis (dysentery) is the presence of blood in patient

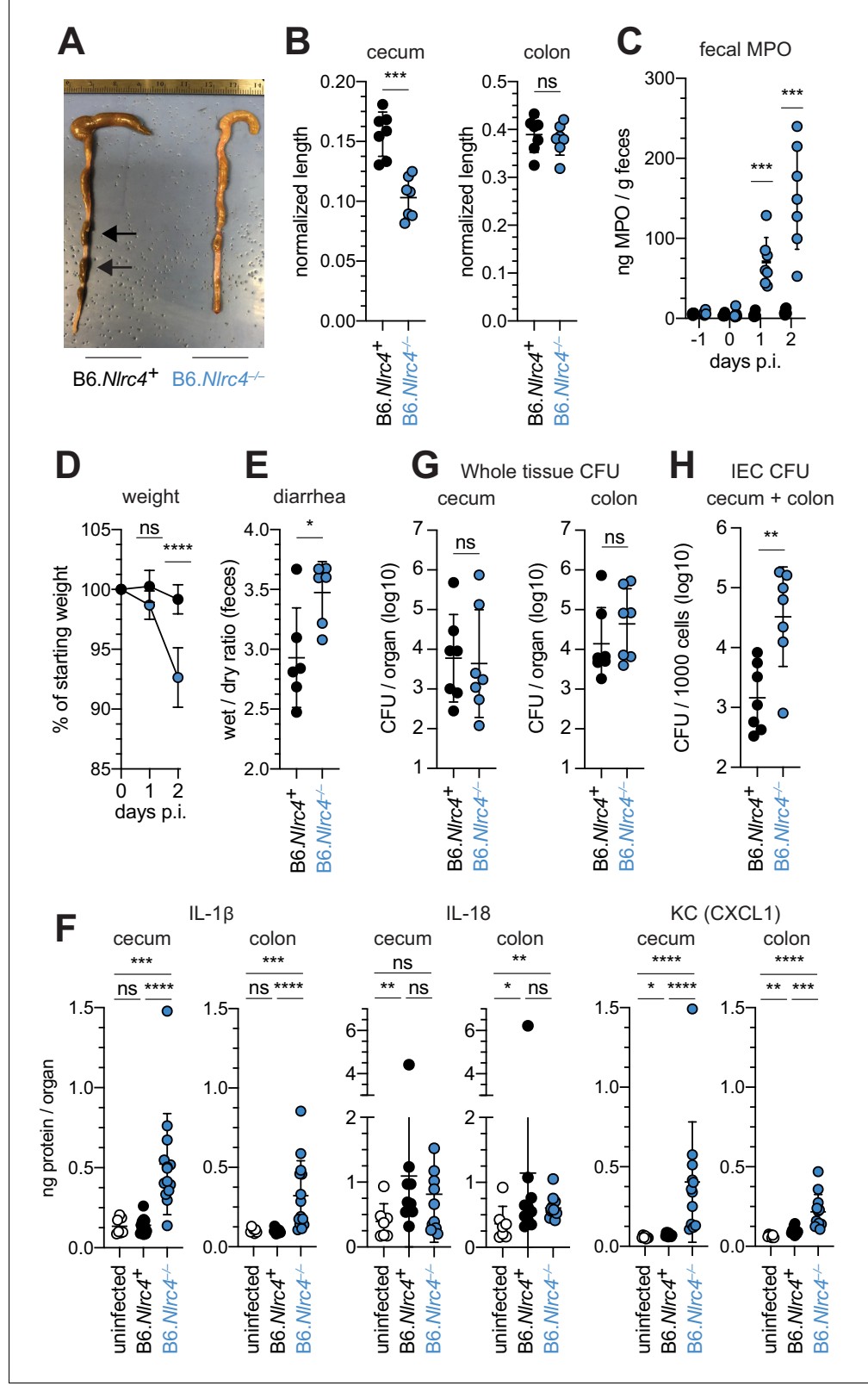

**Figure 3.** *Shigella*-infected B6.*Nlrc4*[−/−] mice exhibit intestinal inflammation and bacterial colonization of IECs. (A–E) B6.*Nlrc4*[+/−] and B6.*Nlrc4*[−/−] littermates were cohoused with B6.WT mice for a minimum of three weeks. Mice were infected with only WT *Shigella* as described for *Figure 2*. Endpoint harvests were performed 48 hr post-infection (p.i.). B6.*Nlrc4*[+/−] and B6.WT mice are collectively referred to as B6.*Nlrc4*[+]. (A) Representative images of *Figure 3 continued on next page*

*Figure 3 continued*

the cecum and colon dissected from B6.*Nlrc4*⁺ and B6.*Nlrc4*⁻/⁻ mice. Note the cecum tissue thickening (size reduction), macroscopic edema, and loose stool (absence of arrows). (**B**) Quantification of cecum and colon lengths. Values were normalized to mouse weight prior to infection; cecum length (cm) / mouse weight (g). (**C**) MPO levels measured by ELISA from feces of B6.*Nlrc4*⁺ and B6.*Nlrc4*⁻/⁻ mice collected -1 through 2 days p.i. (**D**) Mouse weights from 0 through 2 days p.i. Each symbol represents the mean for all mice of the indicated condition. (**E**) Quantification of feces weights before and after dehydration at 2 days p.i. A larger ratio indicates diarrhea. (**F**) IL-1β, IL-18, and KC levels measured by ELISA from tissue of B6.*Nlrc4*⁺ and B6.*Nlrc4*⁻/⁻ mice collected 2 days p.i. (**G**) CFU determination from gentamicin-treated whole tissue homogenates from the cecum or colon of infected mice. (**H**) CFU determination from the IEC enriched fraction of gentamicin-treated cecum and colon tissue (combined). (**B,C,E–H**) Each symbol represents one mouse. Data are representative of three independent experiments. Mean ± SD is shown in (**B-F**). Geometric mean ± SD is shown in (**F, G**). Mann-Whitney test, *p < 0.05, **p < 0.01, ***p < 0.001, ns = not significant (p > 0.05).

The online version of this article includes the following figure supplement(s) for figure 3:

**Figure supplement 1.** Lumenal colonization by *Shigella* is similar between WT and NAIP–NLRC4-deficient mice.

stools. Using an assay to detect occult blood (see Materials and methods), we were unable to detect blood in the feces of NAIP–NLRC4-deficient mice on the B6 background (data not shown). However, when we tested 129.*Nlrc4*⁻/⁻ mouse stools, we found that 4/5 mice infected with WT *Shigella* had occult blood in their feces (*Figure 4I*). In a subsequent infection, 80% (8/10) of 129.*Nlrc4*⁻/⁻ mice had bloody stool (occult blood only, n = 5; macroscopically visible blood, n = 3) (*Figure 4J*). In mice with visible blood, we often observed ruptured blood vessels in the cecum or colon (*Figure 4K*). Thus, 129.*Nlrc4*⁻/⁻ mice recapitulate the bloody stool (dysentery) that is a hallmark of severe human shigellosis.

## Epithelial NLRC4 is sufficient to protect mice from shigellosis

Given the difference in *Shigella* colonization of IECs between WT and NAIP–NLRC4-deficient mice, we next sought to determine if IEC-specific expression of the NAIP–NLRC4 inflammasome is sufficient to protect mice from *Shigella* infection. We thus infected B6 mice that selectively express NLRC4 in IECs. These mice encode a Cre-inducible *Nlrc4* gene on an otherwise *Nlrc4*⁻/⁻ background and are referred to as i*Nlrc4* mice (*Figure 5A*; *Rauch et al., 2017*). Crosses of i*Nlrc4* and *Vil1-Cre* mice generated animals with selective expression of NLRC4 in Villin⁺ IECs. *Shigella* infected *Vil1-Cre*⁺ i*Nlrc4* mice, but not *Cre*⁻ littermate controls, were protected from intestinal inflammation to a similar extent as co-housed *Nlrc4*⁺/⁻ mice (*Figure 5B–F*). Thus, NLRC4 expression in IECs is sufficient to prevent shigellosis.

To further characterize the role of the NAIP–NLRC4 inflammasome during *Shigella* infection of IECs, we generated intestinal epithelial stem cell-derived organoids from the ceca of 129.WT and 129.*Nlrc4*⁻/⁻ mice and established a transwell monolayer infection assay (see Materials and methods). We were unable to recover CFUs from WT IEC monolayers infected with WT *Shigella* (*Figure 6A*). In contrast, 129.*Nlrc4*⁻/⁻ IEC monolayers supported replication of WT *Shigella*. The avirulent non-invasive BS103 strain was detected sporadically at low levels, independent of NAIP–NLRC4, while a strain lacking IcsA, a protein essential for *Shigella* actin tail formation and cell-to-cell spread (*Bernardini et al., 1989*; *Goldberg and Theriot, 1995*), colonized 129.*Nlrc4*⁻/⁻ IEC monolayers to a lesser extent than WT *Shigella*, consistent with loss of IcsA-mediated cell-to-cell spread. Immunostaining for *Shigella* in infected IEC organoid cultures also revealed intracellular replication and actin tail formation (detected by fluorescent phalloidin) exclusively in *Nlrc4*⁻/⁻ IEC monolayers infected with WT *Shigella* (*Figure 6B*). Thus, IEC organotypic infections faithfully recapitulate the NAIP–NLRC4-dependent differences in *Shigella* replication observed *in vivo*. Our results suggest that NAIP–NLRC4 can provide resistance in a cell-type (IEC) intrinsic manner, though additional non-IEC intrinsic functions of NAIP–NLRC4 may also contribute to protection *in vivo*. In addition, our results demonstrate that key *Shigella* virulence factors are functional within mouse cells and can initiate invasion and actin-based motility in mouse IECs, as long as the NAIP–NLRC4 inflammasome is absent.

We sought to determine whether the NAIP–NLRC4 inflammasome prevented *Shigella* colonization of the IEC monolayers by mediating expulsion of the infected cells. Using confocal microscopy,

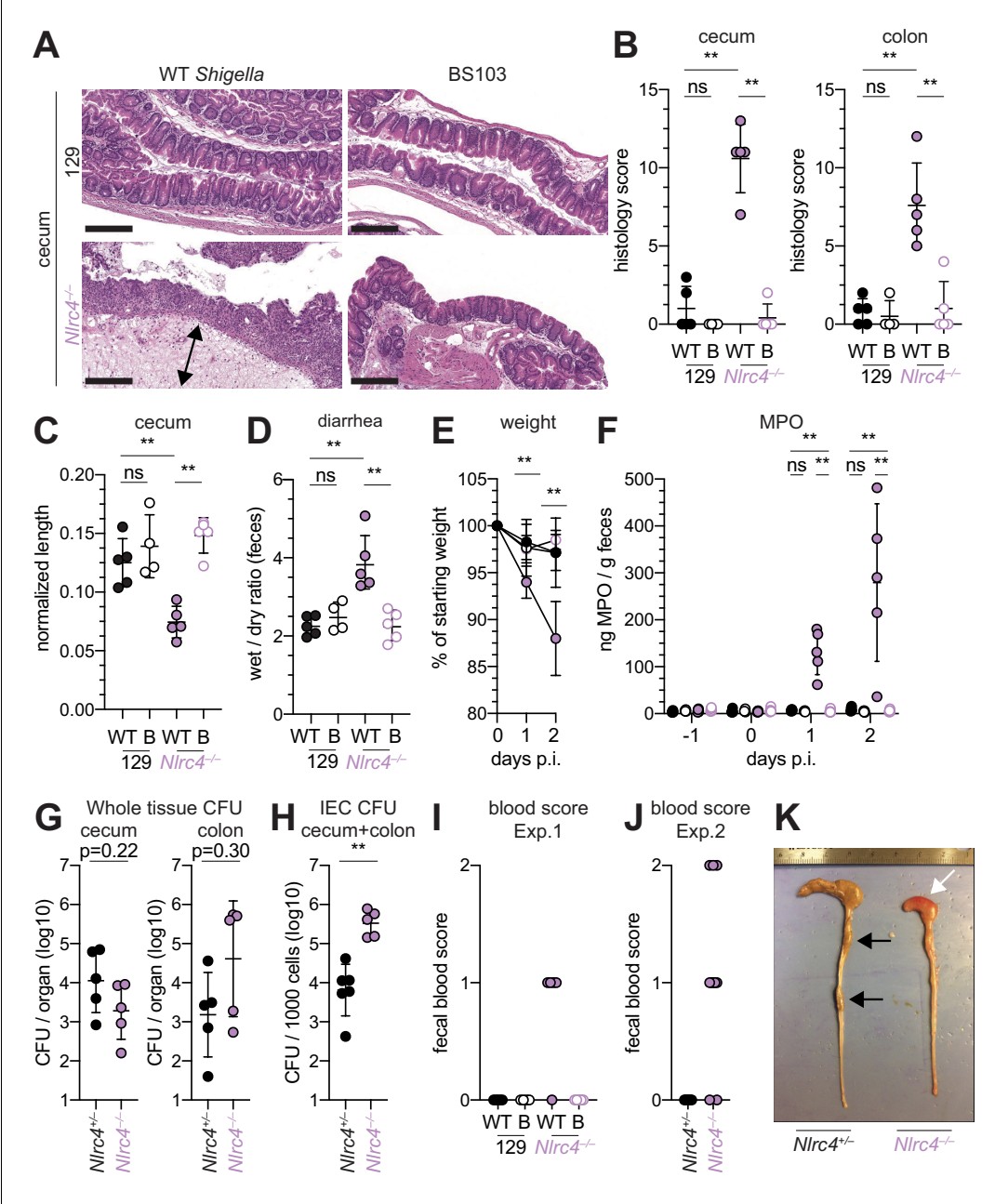

**Figure 4.** *Shigella*-infected 129.*Nlrc4*−/− mice exhibit hallmarks of severe human shigellosis. (**A–H**) 129.*Nlrc4*+/− and 129.*Nlrc4*−/− littermates were infected as described for **Figure 2**. Endpoint harvests were performed at 48 hr post-infection (p.i.). (**A**) Representative images of H&E stained cecum and colon tissue from infected mice. Scale bar, 200 μm. (**B**) Blinded quantification of histology score (cumulative) for tissues in (**A**). Edema, hyperplasia, inflammatory infiltrate, and epithelial cell death were scored from 0 to 4. The final score is the sum of individual scores from each category. (**C**) Quantification of cecum and colon lengths. Values were normalized to mouse weight prior to infection; cecum length (cm) / mouse weight (g). (**D**) Quantification of feces weights before and after dehydration at 2 days p.i. A larger ratio indicates diarrhea. (**E**) Mouse weights at 0 through 2 days p.i. Each symbol represents the mean for all mice of the indicated condition. Statistics refer to both WT *Shigella*-infected 129.*Nlrc4*+/− and 129.*Nlrc4*−/− mice and WT versus BS103 *Shigella*-infected 129.*Nlrc4*−/− mice at both 1 and 2 days p.i. All other comparisons were non-significant. (**F**) MPO levels measured by ELISA from feces of 129.*Nlrc4*+/− and 129.*Nlrc4*−/− mice collected -1 through 2 days p.i. (**G**) CFU determination from gentamicin-treated whole tissue homogenates from the cecum or colon (**H**) CFU determination from the IEC enriched fraction of gentamicin-treated cecum and colon tissue (combined). (**I,J**) Fecal blood scores from feces at two days p.i. 1 = occult blood, 2 = macroscopic blood. (**I** and **J**) show scores from two representative experiments. (**K**) Representative images of the cecum and colon dissected from 129.*Nlrc4*+/− and 129.*Nlrc4*−/− mice. Note the cecum tissue thickening (size reduction), macroscopic edema, and loose stool (absence of arrows), and vascular lesions and bleeding. (**B–D,F–J**) Each symbol represents one

*Figure 4 continued on next page*

*Figure 4 continued*

mouse. Filled symbols, WT *Shigella*; open symbols, BS103. Data are representative of three independent experiments. Mean ± SD is shown in (B, C, D, E, F). Geometric mean ± SD is shown in (G and H). Mann-Whitney test, *p < 0.05, **p < 0.01, ***p < 0.001, ns = not significant (p > 0.05).

The online version of this article includes the following figure supplement(s) for figure 4:

**Figure supplement 1.** Construction and functional characterization of *Nlrc4* knockout mice on the 129S1/SvlmJ genetic background.

we quantified both cell death (the number of PI-positive cells) and cell expulsion (the z-position of PI-positive cells from the monolayer). There were more PI-positive cells in *Shigella*-infected WT versus *Nlrc4*$^{-/-}$ IEC monolayers (*Figure 6C,E*), and the average position of PI-positive cells was significantly above the monolayer (*Figure 6D*). Thus, data from our *ex vivo* IEC monolayer culture system,

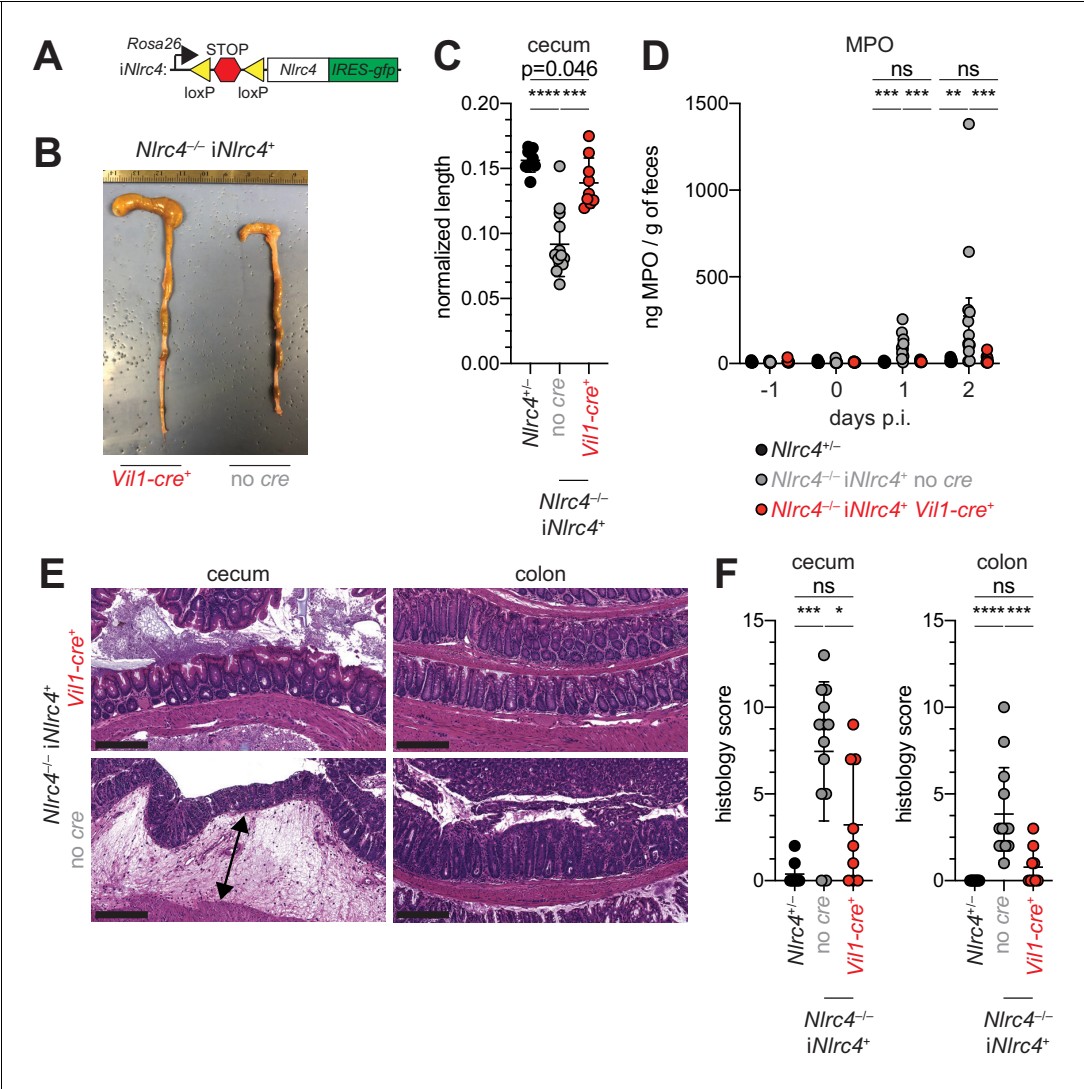

**Figure 5.** NLRC4 expression in IECs is sufficient to prevent shigellosis. (A) Schematic of the B6 *Rosa26* locus containing the i*Nlrc4* cassette, as described previously (*Rauch et al., 2017*). (B–F) *Vil1-cre* positive (+) or negative *Nlrc4*$^{-/-}$ iNlrc4 littermates, or i*Nlrc4*$^{+/-}$ mice were orally infected with 5x10$^7$ CFU of WT *Shigella* 24 hr after oral streptomycin treatment. Endpoint harvests were done 48 hr post-infection (p.i.). (B) Representative images of the cecum and colon dissected from i*Nlrc4 Nlrc4*$^{-/-}$*Vil1-cre* positive or negative mice. (C) Quantification of cecum length reduction normalized to the weight of the animal prior to infection; cecum length (cm) / mouse weight (g). (D) MPO levels measured by ELISA of feces collected -1 through 2 days p.i. (E) Representative images of H&E stained cecum and colon tissue from infected mice. Scale bar, 200 µm. (F) Blinded quantification of histology score (cumulative) for cecum and colon tissue. Data are representative of two independent experiments. Mean ± SD is shown in (C, D, F), Mann-Whitney test, *p < 0.05, **p < 0.01, ***p < 0.001, ns = not significant (p > 0.05). (C,D,F) Each symbol represents one mouse.

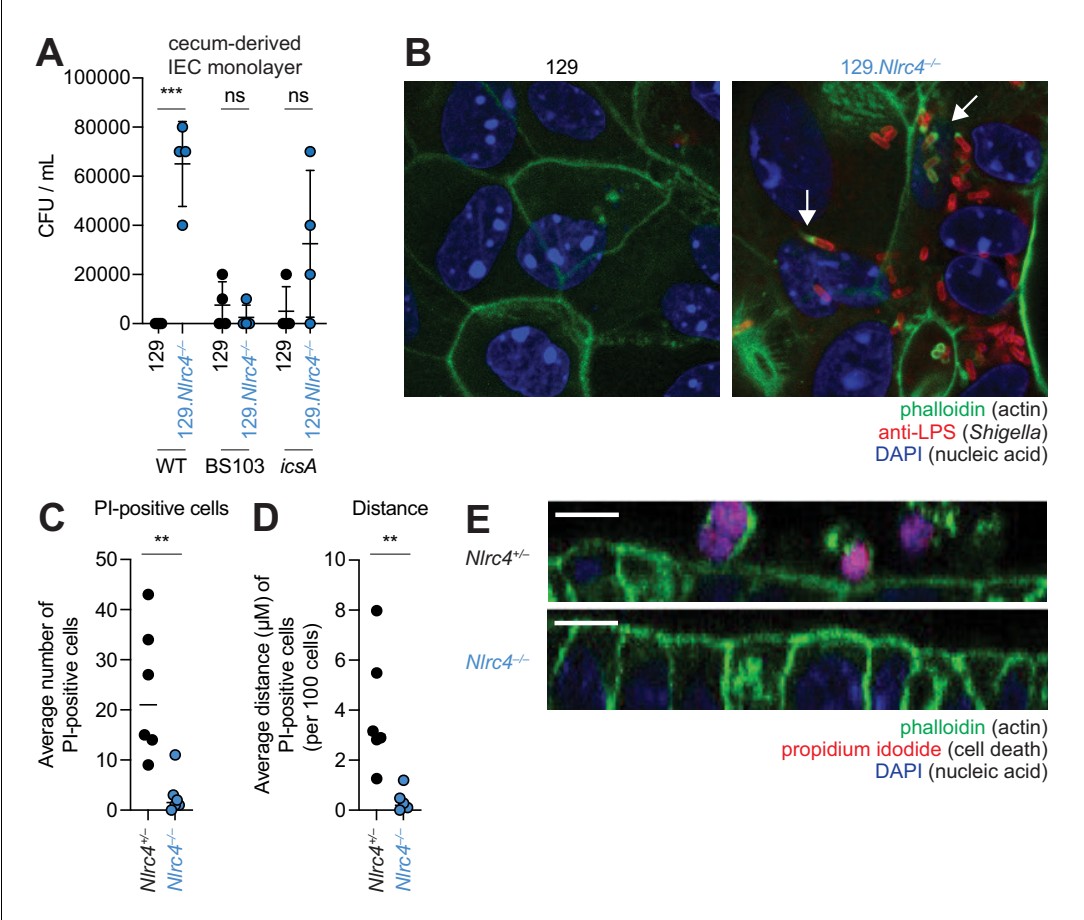

**Figure 6.** NLRC4 prevents *Shigella* colonization and cell expulsion in IEC monolayer cultures. (**A**) *Shigella* (WT, BS103, or *icsA*) CFU from transwell culture of WT or 129.*Nlrc4*<sup>−/−</sup> cecum-derived IEC monolayers. CFU was determined 8 hr p.i. Each symbol represents one infected monolayer. (**B**) Immunofluorescent staining of WT *Shigella*-infected transwell cultures of WT or 129.*Nlrc4*<sup>−/−</sup> cecum-derived IEC monolayers: green, fluorescent phalloidin (actin); red, anti-*Shigella* LPS, blue, DAPI (nucleic acid). (**C,D**) Quantification of the number and position of propidium iodide (PI)-positive cells in *Shigella*-infected 129.*Nlrc4*<sup>+/−</sup> or 129.*Nlrc4*<sup>−/−</sup> cecum-derived IEC monolayers. In (**C**), each symbol represents the average number of PI-positive cells within an imaged field. In (**D**), each symbol represents the average distance of PI-positive cells from the lower boundary of the z-stack, per 100 cells. Two fields were counted for three independent slides. (**E**) A representative XZ-projection of *Shigella*-infected transwell cultures of 129.*Nlrc4*<sup>+/−</sup> or 129.*Nlrc4*<sup>−/−</sup> cecum-derived IEC monolayers showing expulsed PI<sup>+</sup> cells above the monolayer. Green, fluorescent phalloidin (actin); red, PI (cell death); blue, DAPI (nucleic acid). Mann-Whitney test, **p < 0.01, ***p < 0.001, ns = not significant (p > 0.05).

as well as from previous work (*Rauch et al., 2017*), suggest that the NAIP–NLRC4 inflammasome prevents *Shigella* invasion by coordinating the expulsion of infected IECs.

## IcsA-dependent cell-to-cell spread is required for pathogenesis

The *Shigella* IcsA protein is required for virulence in humans (*Collins et al., 2008*; *Mani et al., 2016*; *Orr et al., 2005*). To test if *icsA* is required for pathogenesis in mice, we infected 129.*Nlrc4*<sup>−/−</sup> mice with isogenic WT, *icsA* mutant, or BS103 *Shigella* and monitored disease for 8 days. Mice infected with WT *Shigella* exhibited weight loss (*Figure 7A*), diarrhea (*Figure 7B*), increases in fecal MPO (*Figure 7C*), and blood in their stool (*Figure 7D*). Signs of disease in WT-infected mice peaked between 2 and 3 days post-infection with weight loss, stool consistency, and MPO signal returning to baseline levels at approximately seven days post-infection, consistent with the disease progression and resolution of human shigellosis. Interestingly, 129.*Nlrc4*<sup>−/−</sup> mice infected with *icsA* mutant *Shigella* did not experience weight loss, diarrhea, or fecal blood, and largely phenocopied mice infected with the non-invasive BS103 strain (*Figure 7A–D*). We did observe a slight but significant increase in fecal MPO levels at 1–3 days post-infection in these mice (*Figure 7B*). These results

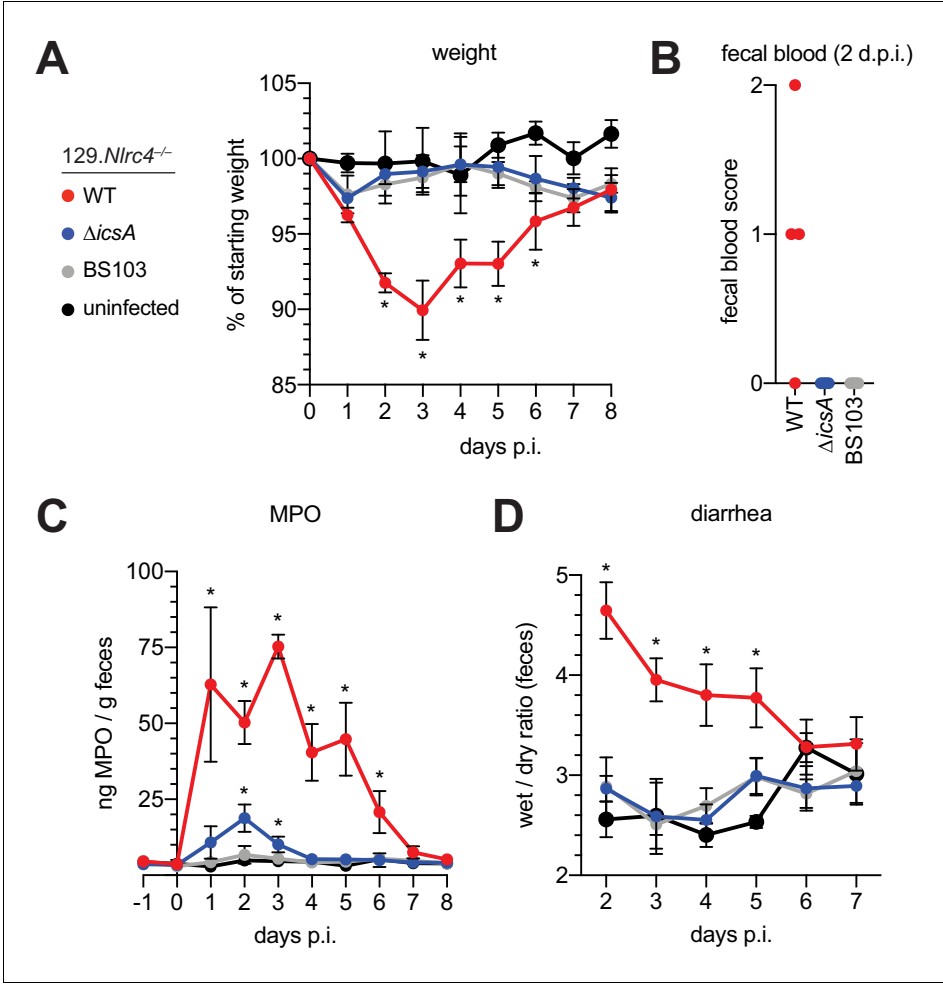

**Figure 7.** 129.*Nlrc4*−/− mice are resistant to attenuated *Shigella* strains. (A–D) 129.*Nlrc4*−/− littermates were uninfected (black) or inoculated orally with 5x10^7 CFU of WT (red), *icsA* mutant (blue), or BS103 (grey) *Shigella* 24 hours after oral streptomycin treatment and monitored for 8 days post-infection (p.i.). (A) Mouse weights. (B) Fecal blood scores from feces at 2 days post-infection (d.p.i.). 1 = occult blood, 2 = macroscopic blood. Each symbol represents feces from one mouse. (C) MPO levels measured by ELISA from feces collected -1 through 8 days p.i. (D) Quantification of diarrhea comparing weight of feces before and after dehydration. A larger ratio indicates diarrhea. (A–C) Each symbol represents the mean at a specific time point for four individual mice per infection condition. Data are representative of two independent experiments. Mean ± SEM is shown in (A–C) Mann-Whitney test, *p < 0.05. In (A,B) significance was determined by independently comparing to Day 0 and to BS103 + uninfected at the same day. In (C), significance was determined by comparing to BS103 and uninfected at the same day.

suggest that, as in humans (*Coster et al., 1999*; *Kotloff et al., 1996*), *icsA* mutants can provoke mild inflammation upon initial colonization of the intestinal epithelium, but that dissemination of bacteria among IECs is a critical driver of severe disease.

## Antibiotic-treated *Nlrc4*−/− mice are susceptible to modest infectious doses

One hallmark of human *Shigella* infection is the low infectious dose relative to other enteric bacterial pathogens. While the reported infectious dose ranges widely from person to person, ingestion of as few as 10–100 bacteria are sufficient to cause symptoms in some humans, although some individuals are resistant to doses of 10^4 bacteria or more (*DuPont et al., 1989*). To determine the infectious dose in our NAIP–NLRC4-deficient mouse model, streptomycin pretreated 129.*Nlrc4*−/− mice were

infected with either $2 \times 10^6$, $2 \times 10^5$, or $2 \times 10^4$ CFU of WT *Shigella* and compared to *Nlrc4*+ littermates infected with $2 \times 10^6$ CFU. Regardless of the inoculum, all mice showed similar levels of lumenal colonization (>$10^8$ CFU/g feces) at one and two days post-infection (*Figure 8A*). This suggests that even a relatively small bacterial population can rapidly expand within the intestines of antibiotic-treated mice. As expected, *Nlrc4*+ mice were largely resistant to disease (*Figure 8B–E*). In contrast, we observed robust disease, including weight loss, cecum shrinkage, elevated fecal MPO, and occult blood in *Nlrc4*−/− mice regardless of the infectious dose (*Figure 8B–E*). The rise in fecal

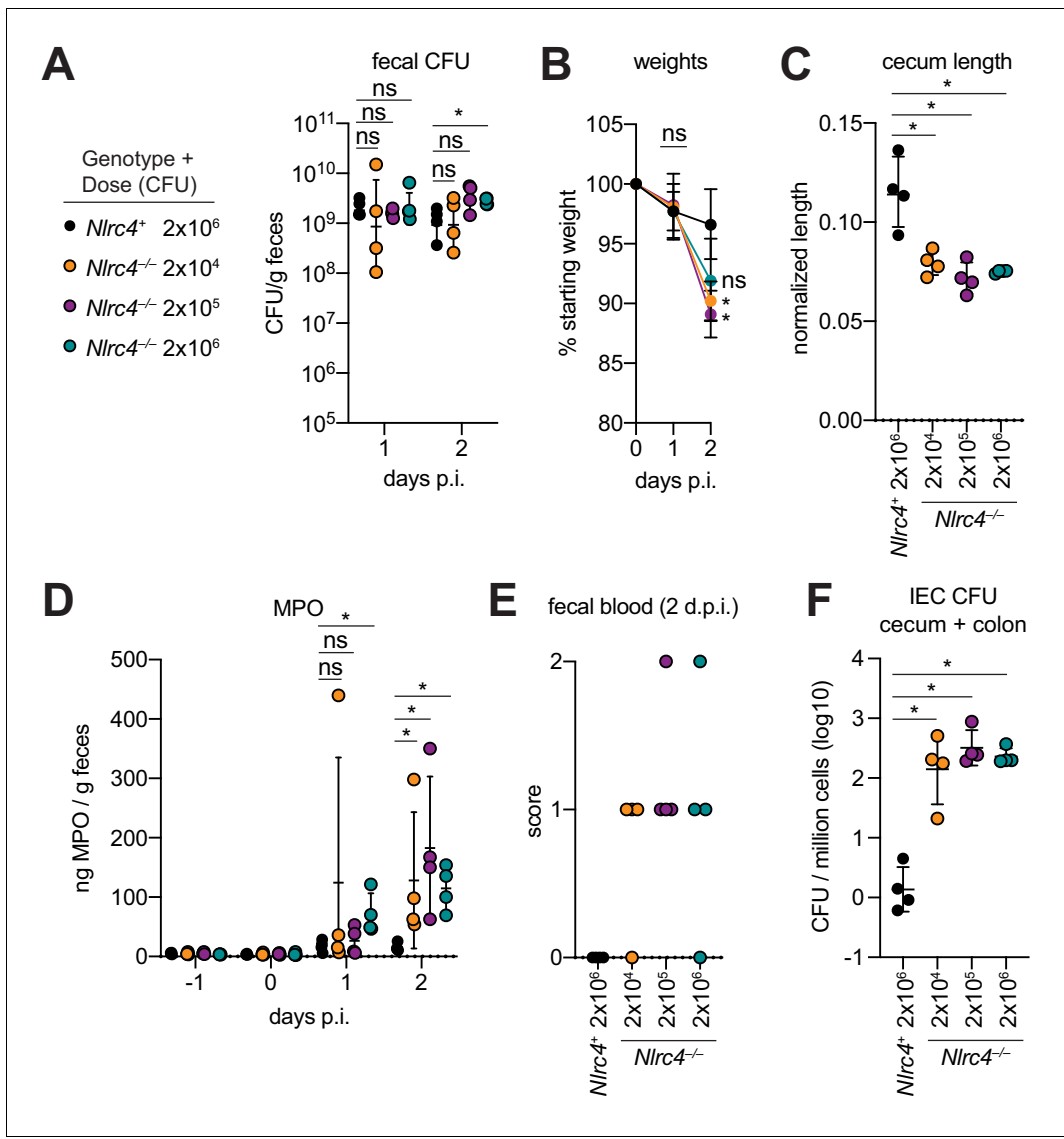

**Figure 8.** Antibiotic treated *Nlrc4*−/− mice are susceptible to modest infectious doses of *Shigella*. (A–F) 129.*Nlrc4*+ mice were inoculated orally with 2x10⁶ (black) and 129.*Nlrc4*−/− littermates were inoculated with 2x10⁶ (teal), 2x10⁵ (purple), or 2x10⁴ (orange) CFU of WT *Shigella* 24 hr after oral streptomycin treatment. Endpoint harvests were done 48 hr post-infection (p.i.). (A) CFU determination from feces. (B) Mouse weights. Each symbol represents the mean at that time point. (C) Quantification of cecum length reduction normalized to the weight of the animal prior to infection; cecum length (cm) / mouse weight (g). (D) MPO levels measured by ELISA from feces collected -1 through 2 days p.i. (E) Fecal blood scores from feces at 2 days post-infection (d.p.i.). 1 = occult blood, 2 = macroscopic blood. (F) CFU determination from the IEC enriched fraction of gentamicin-treated cecum and colon tissue (combined). (A, C–F) Each symbol represents one mouse. Data are representative of two independent experiments. Mean ± SD is shown in (A–D). Geometric mean ± SD is show in (F). Figure Mann-Whitney 8 test, *p < 0.05, ns = not significant (p > 0.05).

MPO in mice receiving smaller inocula was delayed relative to previous infections with $5 \times 10^7$ CFU (*Figure 8D*, *Figure 4F*), suggesting that infectious dose correlates with disease onset. Importantly, there was a 100-fold difference in bacterial colonization of IECs between $Nlrc4^{-/-}$ mice and $Nlrc4^+$ mice, regardless of infectious dose, but no difference in IEC colonization among $Nlrc4^{-/-}$ mice inoculated with $2 \times 10^4 - 2 \times 10^6$ CFU (*Figure 8F*). These results indicate that in antibiotic treated mice, disease is fully penetrant at $2 \times 10^4$ CFU, a dose that recapitulates the infectious dose observed in at least some human patients.

## Discussion

Here, we demonstrate that the NAIP–NLRC4 inflammasome is a formidable species-specific barrier to *Shigella* invasion of the intestinal epithelium. *Shigella* infection of antibiotic pre-treated, NAIP–NLRC4-deficient mice recapitulates key features of human shigellosis, including bacterial invasion of and replication in IECs, severe inflammatory disease at relevant sites (e.g. colon, cecum), and bloody diarrhea. While ocular, pulmonary, and intraperitoneal *Shigella* infections have been used to assess bacterial virulence in mice, these models do not feature bacterial colonization of intestinal epithelial cells, a key event required for pathogenesis in humans. Thus, inflammasome-deficient mice provide the first physiologically relevant mouse model of bacillary dysentery. The genetic and immunological tractability of this system relative to other small animal oral infection models (rabbit, guinea pigs) should pave the way for detailed genetic and mechanistic *in vivo* studies of the host factors underlying *Shigella* pathogenesis that have long been elusive.

There remain key differences between orogastric *Shigella* infection of NAIP–NLRC4-deficient mice and human infection. First, even genetically susceptible mice require oral antibiotic treatment to alter the mouse microbiome and allow *Shigella* to establish a replicative niche in the gut. However, this limitation is not unique to our *Shigella* model, as antibiotic pretreatment is a common practice in many mouse models of enteric bacterial infections (*Barthel et al., 2003*; *Becattini et al., 2017*; *Bou Ghanem et al., 2013*). Interestingly, *S. sonnei* encodes a type VI secretion system that mediates competition with other bacterial species (*Anderson et al., 2017*). This finding suggests that interbacterial interactions may also be an important selective pressure acting on *Shigella* in the human gut. An important next step will be to identify interactions between *Shigella* and the human versus mouse microbiome that define the species-specific differences in colonization resistance.

In our model, we were able to achieve shigellosis in mice inoculated with as few as $2 \times 10^4$ CFU. Although much is made of the low infectious dose of *Shigella* required to cause disease in humans, we note that experimental human infection studies only variably achieved disease with 100 CFU (39%) (*DuPont et al., 1969*) and even higher inocula failed to cause disease in some individuals $10^5$–$10^8$ CFU (64%) (*DuPont et al., 1989*). We also note that *Shigella* rapidly expands in the antibiotic-treated mouse gut regardless of starting inoculum, suggesting that there is a threshold dose required to establish infection, beyond which a larger inoculum may have little significance to the disease outcome. Ultimately, infectious dose during natural infections likely varies depending on a multitude of environmental and host factors. While many questions remain, our model now provides a means by which to dissect these key questions, which will allow future refinement and greater utility of the model to understand human infection and disease.

A long-held belief is that *Shigella* exploits inflammasomes to induce pyroptosis. Pyroptotic cell death is presumed to allow bacteria to escape macrophages and invade the basolateral surface of polarized enterocytes (*Ashida et al., 2014*; *Lamkanfi and Dixit, 2010*; *Schnupf and Sansonetti, 2019*). Although we do not directly address this possibility, our experiments suggest that *Shigella* has instead evolved to inhibit or evade the human NAIP–NLRC4 inflammasome to limit intestinal epithelial cell death. Indeed, we find that the NAIP–NLRC4 inflammasome plays a critical role in host defense by restricting *Shigella* replication and spread in IECs. Further supporting this notion, only $Nlrc4^{-/-}$ but not WT IEC organoid monolayers are permissive to *Shigella* infection. Our data are consistent with the NAIP–NLRC4 inflammasome providing defense by coordinating the expulsion of *Shigella*-infected cells. Similarly, *Salmonella* infected IECs are expelled from the intestinal epithelial barrier in an NLRC4-dependent manner (*Rauch et al., 2017*; *Sellin et al., 2014*). Thus, epithelial inflammasomes coordinate the expulsion of infected IECs as a general defense strategy against enteric bacterial pathogens. A non-mutually exclusive possibility is that NAIP–NLRC4 (or other inflammasomes) are required for *Shigella* invasion and or pathogenesis in non-IECs, including

macrophages. Interestingly, in *Nlrc4*$^{-/-}$ mice, IL-1β and KC levels in target tissues are elevated, indicating that *Shigella* may encounter multiple inflammasomes at distinct stages of infection with opposing consequences for bacterial replication and host pathogenesis.

We only observe bloody diarrhea in *Nlrc4*$^{-/-}$ mice generated on the 129 genetic background. 129 mice naturally harbor a null *Casp11* allele (*Kayagaki et al., 2011*). Although other genetic differences may contribute to the variation in susceptibility to *Shigella* between B6 and 129 NAIP–NLRC4-deficient mice, we speculate that both the NAIP–NLRC4 and the CASP11 inflammasomes mediate protection in IECs against *Shigella* invasion. However, the susceptibility of B6 NAIP–NLRC4-deficient (but CASP11$^+$) mice, as well as the resistance of *Casp1/11*$^{-/-}$ mice, suggest that these inflammasomes are not strictly redundant and that NAIP–NLRC4 alone is sufficient to confer resistance to shigellosis in mice. The *Shigella* effector OspC3 antagonizes the human LPS sensor CASP4 (*Kobayashi et al., 2013*), and IEC-expressed CASP4 provides protection against other human bacterial pathogens (*Holly et al., 2020*; *Knodler et al., 2014*), underscoring the importance of the LPS-sensing pathway during human infection. Similarly, our finding that *Shigella* appears to suppress the human NAIP–NLRC4 inflammasome implies that evasion and/or antagonism of inflammasomes is a general virulence strategy during human infections.

There is currently no licensed *Shigella* vaccine, and very limited knowledge of what vaccine-induced immune responses would be desirable to elicit to mediate protection (*Barry et al., 2013*; *Mani et al., 2016*). Our new shigellosis model will finally allow the field to leverage the outstanding genetic and immunological tools and reagents in the mouse to address fundamental questions about the immune response to *Shigella*. Our finding that NAIP–NLRC4 inflammasome-deficient mice clear the attenuated *Shigella icsA* strain, derivatives of which are currently deployed in human vaccine trials (*Collins et al., 2008*; *Coster et al., 1999*; *Ranallo et al., 2014*), speaks to the readiness of our model to support testing and development of *Shigella* therapeutics. More broadly, our results also provide a striking example of how inflammasomes provide an important species-specific barrier against infection.

# Materials and methods

**Key resources table**

| Reagent type (species) or resource | Designation | Source or reference | Identifiers | Additional information |
|---|---|---|---|---|
| Strain, strain background (*Mus musculus*, C57BL/6J) | WT | Jax and Vance Lab colony, Jax stock No. 000664 | | Source of BMDMs |
| Strain, strain background (*Mus musculus*, C57BL/6J) | *Nlrc4*$^{-/-}$ | Vance Lab colony, (*Tenthorey et al., 2020*) | | Source of BMDMs |
| Strain, strain background (*Mus musculus*, C57BL/6J) | Δ*Naip* mice (also called *Naip1-6*$^{Δ/Δ}$) | Vance Lab colony (*Rauch et al., 2016*) | | |
| Strain, strain background (*Mus musculus*, C57BL/6J) | *Casp1/11*$^{-/-}$ | Vance Lab colony (*Li et al., 1995*) | | |
| Strain, strain background (*Mus musculus*, C57BL/6J and C57BL/6N mixed) | *iNlrc4* | Vance Lab colony (*Rauch et al., 2017*) | | These mice encode a Cre-inducible *Nlrc4* gene in the *Rosa26* locus. |
| Strain, strain background (*Mus musculus*, C57BL/6J) | *Vil1-Cre* | Vance Lab colony, Jax stock No. 004586 | | |

*Continued on next page*

*Continued*

| Reagent type (species) or resource | Designation | Source or reference | Identifiers | Additional information |
|---|---|---|---|---|
| Strain, strain background (*Mus musculus*, 129S1/SvImJ) | WT | Jax and Vance Lab colony, Jax stock No. 002448 | | Source of BMDMs, IECs |
| Strain, strain background (*Mus musculus*, 129S1/SvImJ) | *Nlrc4$^{-/-}$* | Vance Lab colony; this paper | | Source of BMDMs, IECs; see Materials and methods, 'Animal Procedures' |
| Cell line (*Homo-sapiens*) | THP-1, monocyte-like (WT) | Gift from Hornung Lab | | |
| Cell line (*Homo-sapiens*) | THP-1, monocyte-like (*Nlrc4$^{-/-}$*) | Gift from Hornung Lab | | |
| Cell line (*Homo-sapiens*) | 293T, embryonic kidney | Berkeley Cell Culture Facility | | |
| Antibody | Polyclonal Goat anti human IL-1β | R and D | AF-201-NA | (1:1000) |
| Antibody | Monoclonal mouse anti *Aequorea victoria* GFP | Clontech | JL-8 | (1:1000) |
| Antibody | Monoclonal mouse anti rat b-TUBULIN | Sigma | clone TUB 2.1 | (1:1000) |
| Antibody | Rabbit polyclonal anti *Shigella* | Abcam | ab65282 | (1:100) |
| peptide, recombinant protein | 488-phalloidin | Cytoskeleton Inc | PHDG1-A | (1:500) |
| Antibody | anti-IL-1β capture and detection | R and D | DY401 | (concentration not indicated by manufacturer) |
| Antibody | Anti-MPO capture and detection | R and D | DY3667 | (concentration not indicated by manufacturer) |
| Antibody | Anti-KC capture and detection | R and D | DY453 | (concentration not indicated by manufacturer) |
| Antibody | Capture: moncloncal rat anti mouse IL-18 | BD Biosciences and eBioscience | catalog # D047-3, clone 74 | 1:1000 |
| Antibody | Detection: moncloncal rat anti mouse IL-18 | BD Biosciences and eBioscience | catalog # D048-6, clone 93–10 c | 1:2000 |
| Chemical compound, drug | Rock inhibitor Y27632 | Stem Cell | 72304 | |

*Continued on next page*

*Continued*

| Reagent type (species) or resource | Designation | Source or reference | Identifiers | Additional information |
|---|---|---|---|---|
| Chemical compound, drug | TGF-Smad inhibitor SB431542 | Stem Cell | 72234 | |
| Peptide, recombinant protein | Mouse TRANCE (RANKL) (carrier-free) | BioLegend | 577102 | |
| Chemical compound, drug | Basement Membrane Matrix (Matrigel) | Fisher | CB-40234A | |
| Peptide, recombinant protein | LFn-PA | This paper | | see Materials and methods, 'Toxins' |
| Peptide, recombinant protein | LFn-MxiH | This paper | | see Materials and methods, 'Toxins' |
| Peptide, recombinant protein | LFn-FlaA | *Rauch et al., 2017* | | |
| Strain, strain background (*Shigella flexneri* serovar 2a) | WT 2457T | | | |
| Strain, strain background (*Shigella flexneri* serovar 2a) | BS103 | *Maurelli et al., 1984* | | |
| Strain, strain background (*Shigella flexneri* serovar 2a) | icsA | *Goldberg and Theriot, 1995* | | |

## Cell culture

293T cells were cultured in DMEM supplemented with 10% FBS and 2 mM L-glutamine. THP1 cells were cultured in RPMI supplemented with 10% FBS and 2 mM L-glutamine. B6 primary BMMs were cultured in RPMI supplemented with 10% FBS, 5% 3T3-MCSF supernatant, 100 U/ml penicillin, 100 mg/ml streptomycin and 2 mM L-glutamine. THP1 cells were a gift from Veit Hornung, and generated as previously described (*Gaidt et al., 2017*). Cells were grown in media without antibiotics for infection experiments. All cell lines tested negative for mycoplasma.

## Bacterial strains

All experiments were conducted with the *S. flexneri* serovar 2a WT 2457T strain, or the WT-derived virulence plasmid-cured strain BS103 (*Maurelli et al., 1984*) or *icsA* mutant (*Goldberg and Theriot, 1995*; *Makino et al., 1986*). The *icsA* mutant strain was a gift from Marcia Goldberg. Natural streptomycin resistant strains of 2457T and BS103 were generated by plating cultured bacteria on tryptic soy broth (TSB) plates containing 0.01% Congo Red (CR) and increasing concentrations of streptomycin sulfate. Streptomycin-resistant strains were confirmed to grow indistinguishably from parental strains in TSB broth lacking antibiotics, indicating an absence of streptomycin-dependence.

## Toxins

Recombinant proteins for cytosolic delivery of *Shigella* MxiH were produced using the BD Baculo-GOLD system for protein expression in insect cells. The MxiH coding sequence was subcloned into pAcSG2-6xHIS-LFn using the primers: PSMpr943 F (BamHI) 5' - GAAAGG GGATCC ATG AGT GTT

ACA GTA CCG GAT AAA GAT TGG ACT CTG - 3' and PSMpr944 R (NotI) 5' - GAAAGG GCGGCCGC TTA TCT GAA GTT TTG AAT AAT TGC AGC ATC AAC ATC C - 3'. The PA-6xHIS coding sequence was subcloned from pET22b-PA-6xHIS (*Rauch et al., 2016*) into pAcSG2 using the primers: PSMpr896 F (XhoI) 5' - GAAAGG CTCGAG ATG GAA GTT AAA CAG GAG AAC CGG TTA TTA AAT GAA TC - 3' and PSMpr897 R (NotI) 5' - GAAAGG GCGGCCGC TCA GTG GTG GTG GTG GTG GTG T - 3'. Constructs were co-transfected with BestBac linearized baculovirus DNA (Expression Systems) into SF9 cells following the manufacturer's protocol to generate infectious baculovirus. Primary virus was amplified in SF9 cells. Recombinant proteins were produced by infecting 2L of High Five cells with 1 ml of amplified virus/L cells. Cells were harvested ~ 60 hr after infection by centrifugation at 500x$g$ for 15 min. Cell pellets were resuspended in lysis buffer (50 mM Tris pH7.4, 150 mM NaCl, 1% NP-40 with protease inhibitors) and lysed on ice using a dounce homogenizer. Samples were then clarified at 24,000x$g$ for 30 min and supernatants were batch bound to 1 ml nickel resin for 2 hr at 4°C. Samples were column purified by gravity. Resin was washed with 100 ml of wash buffer (20 mM Tris pH7.4, 400 mM NaCl, 20 mM imidizole). Sample was eluted with 1 ml fractions of elution buffer (20 mM Tris pH7.4, 150 mM NaCl, 250 mM imidizole). Peak elutions were pooled and buffer exchanged into 20 mM Tris pH7.4.

## Infection of cells in culture

*S. flexneri* was grown at 37°C on tryptic soy agar plates containing 0.01% Congo red (CR), supplemented with 100 µg/ml spectinomycin and 100 µg/ml carbenicillin for growth of the *icsA* strain. For infections, a single CR-positive colony was inoculated into 5 ml TSB and grown shaking overnight at 37°C. Saturated cultures were back-diluted 1:100 in 5 ml fresh TSB shaking for ~ 2 hr at 37°C. THP1 and BMM cells were seeded at 100,000 cells per well of a Nunc F96 MicroWell white polystyrene plate. Bacteria were washed three times in cell culture media, then spinfected onto cells for 10 min at 500x$g$. Bacterial invasion was allowed to proceed for an additional 20 min at 37°C, followed by three washes in with cell culture media containing 25 mg/ml gentamicin. Cells were then maintained in cell culture media containing 2.5 mg/ml gentamicin with propidium iodide (Sigma, diluted 1:100 from stock) at 37°C for the duration of the assay (30 min to 4 hr). A MOI of 10 was used unless otherwise specified. For suppression assays, PMA-differentiated THP1 cells were infected as described above for 1 hr. Media was then replaced with cell culture media containing 2.5 mg/ml gentamicin and propidium iodide (1:100) and either 10 µg/ml PA only, PA with 1.0 µg/ml LFn-MxiH, or 10 µM nigericin. PI uptake was measured using a SpectraMax M2 plate reader, and 100% cell death was set by normalizing values of infected wells to cells lysed with 1% Triton X-100 after background subtraction based on media only controls.

## Establishment, propagation, and infection of IECs

Primary intestinal epithelial stem-cell-derived organoids from the cecum were isolated and maintained in culture as previously described (*Miyoshi and Stappenbeck, 2013*). Each transwell monolayer culture was established 1:1 from a confluent enteroid Matrigel (Corning, 356255) 'dome.' Enteroids were disassociated from Matrigel with 0.25% trypsin for 10 min, manually disrupted, resuspended in monolayer culture wash media (ADMEM/F12 supplemented with 20% FBS, 1% L-glutamine) and plated on polycarbonate transwells (Corning, 3413) that had been pre-coated for > 1 hr at 37°C with 1:30 Matrigel:wash media. Monolayer cultures were differentiated for 12–14 days in complete monolayer culture media (monolayer culture wash media mixed 1:1 with LWRN-conditioned media supplemented with 10 µM Y27632 (Stem Cell), and RANKL (BioLegend) in the absence of SB431542). Two days prior to infection, cells were cultured in antibiotic-free monolayer culture media. Monolayers were treated with 20 µM EGTA for 15 min prior to *Shigella* infection (MOI = 10). Bacterial invasion was allowed to proceed for 2 hr, followed by gentamicin washes as described above to both upper and lower compartments, then maintained in monolayer culture media containing 2.5 mg/ml gentamicin with propidium iodide (Sigma, diluted 1:100 from stock) at 37°C for the duration of the assay (1 hr for IF and 8 hr for CFU determination). For IF, cells were washed in PBS, fixed in 4% paraformaldehyde for 15 min, permeabilized in 0.1% Triton X-100 for 15 min, blocked in PBS with 2% BSA, 0.1% Tween-20 for 1 hr. Primary antibodies were incubated overnight, followed by a 1 hr incubation with fluorophore-conjugated secondary antibodies and a 10-min incubation with DAPI and fluorophore-conjugated phalloidin. Slides were analyzed on a Zeiss LSM710. The number

and z-axis position of PI-positive cells was performed using Imaris. Reagents for immunofluorescence: anti-*Shigella* (Abcam, ab65282), 488 phalloidin (PHDG1-A, Cytoskeleton Inc), Alexfluor conjugated secondary antibodies (Invitrogen). To determine bacterial replication in IECs, 8 hr post-infection monolayers were washed three times with PBS, lysed in 1% Triton X-100, and bacteria were plated for CFU determination.

## Reconstituted NAIP–NLRC4 inflammasome activity assays

To reconstitute inflammasome activity in 293T cells, constructs (100 ng of each) producing human NAIP, NLRC4, CASP1, and IL-1β were co-transfected with constructs (200 ng of each) producing *Shigella* MxiI, MxiH or empty vector (pcDNA3) using Lipofectamine 2000 (Invitrogen) following the manufacturer's protocol and harvested 24 hr post-transfection. For experiments using recombinant proteins, fresh media containing 10 µg/ml PA and 1.0 µg/ml LFn-MxiH was added to cells for 3–4 hr. In all experiments, cells were lysed in RIPA buffer with protease inhibitor cocktail (Roche).

## Immunoblot and antibodies

Lysates were clarified by spinning at 16,100x*g* for 10 min at 4°C. Clarified lysates were denatured in SDS loading buffer. Samples were separated on NuPAGE Bis-Tris 4–12% gradient gels (Thermo-Fisher) following the manufacturer's protocol. Proteins were transferred onto Immobilon-FL PVDF membranes at 375mA for 90 min and blocked with Odyssey blocking buffer (Li-Cor). Proteins were detected on a Li-Cor Odyssey Blot Imager using the following primary and secondary antibodies: anti-IL-1β (R and D systems, AF-201-NA), anti-GFP (Clontech, JL8), anti-TUBULIN (Sigma, clone TUB 2.1), Alexfluor-680 conjugated secondary antibodies (Invitrogen).

## Animal procedures

All mice were maintained in a specific pathogen free colony until 1–2 weeks prior to infection, maintained under a 12 hr light-dark cycle (7 am to 7 pm), and given a standard chow diet (Harlan irradiated laboratory animal diet) *ad libitum*. Wild-type C57BL/6J and 129S1/SvImJ mice were originally obtained from the Jackson Laboratories. 129.*Nlrc4*−/− animals were generated by targeting *Nlrc4* via CRISPR-Cas9 mutagenesis. CRISPR/Cas9 targeting was performed by injection of Cas9 mRNA and sgRNA into fertilized zygotes, essentially as described previously (*Wang et al., 2013*). Founder mice were genotyped by PCR and sequencing using the primers: JLR035 F 5' CAGGTCACAGAAG AAGACCTGAATG 3' and JLR036 R 5' CACCTGGACTCCTGGATTTGG 3'. Founders carrying mutations were bred one generation to wild-type mice to separate modified haplotypes. Homozygous lines were generated by interbreeding heterozygotes carrying matched haplotypes. B6.Δ*Naip* mice were generated as described previously (*Rauch et al., 2016*). B6.*Nlrc4*−/− (*Tenthorey et al., 2020*) and i*Nlrc4* (*Rauch et al., 2017*) mice were previously described. i*Nlrc4* mice were crossed to the *Nlrc4*−/− line and then further crossed to *Vil1*-cre (Jax strain 004586) transgenic lines on a *Nlrc4*−/− background. Animals used in infection experiments were littermates or, if not possible, were co-housed upon weaning. In rare cases when mice were not co-housed upon weaning, mice were co-housed for at least one week prior to infection. Animals were transferred from a SPF colony to an ABSL2 facility at least one weeks prior to infection. All animal experiments complied with the regulatory standards of, and were approved by, the University of California, Berkeley Animal Care and Use Committee.

## *In vivo Shigella* infections

Mouse infections were performed in 6–16 week old mice. Initially, mice deprived of food and water for 4–6 hr were orally gavaged with 100 µL of 250 mg/mL streptomycin sulfate dissolved in water (25 mg/mouse) and placed in a cage with fresh bedding. 24 hr later, mice again deprived of food and water for 4–6 hr were orally gavaged with 100 µL of log-phase, streptomycin resistant *Shigella flexneri* 2457T, BS103, or *icsA* mutant at the indicated dose, prepared as above and resuspended in PBS. Mouse weights and fecal pellets were recorded or collected daily from 1 day prior to infection to the day of euthanasia and harvest (usually 2 days post-infection) to assess the severity of disease and biomarkers of inflammation. Infection inputs were determined by serially diluting a fraction of the initial inoculum and plating on TSB plates containing 0.01% CR and 100 µg/mL streptomycin.

### Fecal CFUs, fecal MPO ELISAs, wet/dry ratio, fecal occult blood

Fecal pellets were collected in 2 mL tubes, suspended in 2% FBS in 1 mL of PBS containing protease inhibitors, and homogenized. For CFU enumeration, serial dilutions were made in PBS and plated on TSB plates containing 0.01% CR and 100 µg/mL streptomycin sulfate. For MPO ELISAs, fecal homogenates were spun at 2,000 g and supernatants were plated in triplicate on absorbent immunoassay 96-well plates. Recombinant mouse MPO standard, MPO capture antibody, and MPO sandwich antibody were purchased from R&D Systems. Wet/dry ratios were determined by weighing fecal pellets before and after they had been dried in a fume hood. The presence or absence of fecal occult blood in fresh pellets was determined using a Hemoccult blood testing kit (Beckman Coulter).

### Tissue ELISAs

The cecum and colon were isolated from mice at two days post infection and rinsed 5x in PBS to removal fecal contents. Organs were then homogenized in 1 mL of 2% FBS in PBS + protease inhibitors, spun down at 2000 g, and supernatants were plated in duplicate on absorbent immunoassay 96-well plates. Recombinant mouse IL-1β and KC standards, capture antibodies, and sandwich antibodies were purchased from R and D. IL-18 paired antibodies were purchased from BD Biosciences and eBioscience and recombinant IL-18 standard from eBiosciene.

### Histology

Mice were euthanized at two days post-infection by $CO_2$ inhalation and cervical dislocation. Ceca and colons from mice were isolated, cut longitudinally, removed of lumenal contents, swiss-rolled, and fixed in methacarn followed by transfer to 70% ethanol. Samples were processed by routine histologic methods on an automated tissue processor (TissueTek, Sakura), embedded in paraffin, sectioned at 4 µm thickness on a rotary microtome, and mounted on glass slides. Sections were stained with hematoxylin and eosin on an automated histostainer and coverslipped. Histopathological evaluation was performed by light microscopy (Olympus BX45, Olympus Corporation) at magnifications ranging from x20 to x600 by a board-certified veterinary pathologist (I.L.B.) who was blinded to the experimental groups at the time of evaluation. Representative images were generated as Tiff files from digitized histology slides scanned on a digital slide scanner (Leica Aperio AT2, Leica Biosystems). Images were taken using freely downloadable software (Image Scope, Leica Aperio, Leica Biosystems) and processed in Adobe Photoshop. Photo processing was confined to global adjustments of image size, white balance, contrast, brightness, sharpness, or correction of lens distortion and did not alter the interpretation of the image. Sample preparation, imaging, and histology scoring was conducted by the Unit for Laboratory Animal Medicine at the University of Michigan.

### Intestinal CFU determination

To enumerate whole tissue intestinal CFU, ceca and colons from mice were isolated, cut longitudinally and removed of lumenal contents, placed in culture tubes containing 400 µg/mL gentamicin antibiotic in PBS, vortexed, and incubated in this solution for 1–2 hr. Organs were washed five times in PBS to dilute the gentamicin, homogenized in 1 mL of PBS, serially diluted, and plated on TSB agar plates containing 0.01% CR and 100 µg/mL streptomycin. To enumerate intracellular CFU from the intestinal epithelial cell fraction of the cecum and colon, organs prepared as above were incubated in RPMI with 5% FBS, 2 mM L-glutamine, and 400 µg/ml of gentamicin for 1–2 hr. Tissues were then washed five times in PBS, cut into 1 cm pieces, placed in 15 mL of stripping solution (HBSS, 10 mM HEPES, 1 mM DTT, 2.6 mM EDTA), and incubated at 37°C for 25 min with gentle agitation. Supernatants were passed through a 100 µm filter and the remaining pieces of tissue were shaken in a 50 mL conical with 10 mL of PBS and passed again through the 100 µm filter. This enriched epithelial cell fraction was incubated in 50 µg/mL gentamicin for 25 min on ice, spun at 300x*g* at 4°C for 8 min, and washed twice by aspirating the supernatant, resuspending in PBS, and spinning at 300x*g* at 4°C for 5 min. After the first wash, a fraction of cells were set aside to determine the cell count. After the second wash, the pellet was resuspended and lysed in 1 mL of 1% Triton X-100. Serial dilutions were made from this solution and plated on TSB agar plates containing 0.01% CR and 100 µg/ml streptomycin and CR+ positive colonies were counted following overnight incubation at 37°C.

## Acknowledgements

We thank M Goldberg for advice and for sharing the *icsA* mutant *Shigella* strain. We are grateful to G Barton and H Darwin for comments on the manuscript, and members of the Vance and Barton Labs for discussions. Funding: REV is an HHMI Investigator and is supported by NIH AI075039 and AI063302; PSM is supported by a Jane Coffin Childs Memorial Fund postdoctoral fellowship. JLR is an Irving H Wiesenfeld CEND Fellow; EAT is supported by the UC Berkeley Department of Molecular and Cell Biology NIH Training Grant 5T32GM007232-42; CFL is a Brit d'Arbeloff MGH Research Scholar and supported by NIH AI064285 and NIH AI128743; IR is supported by the Medical Research Foundation MRF2012.

## Additional information

### Competing interests

Russell E Vance: Reviewing editor, *eLife*. The other authors declare that no competing interests exist.

### Funding

| Funder | Grant reference number | Author |
| --- | --- | --- |
| Howard Hughes Medical Institute | | Russell E Vance |
| National Institutes of Health | AI075039 | Russell E Vance |
| National Institutes of Health | AI064285 | Cammie F Lesser |
| Jane Coffin Childs Memorial Fund for Medical Research | Postdoctoral Fellowship | Patrick S Mitchell |
| Irving H. Wiesenfeld CEND Fellow | Graduate Student Fellowship | Justin L Roncaioli |
| UC Berkeley Department of Molecular and Cell Biology, NIH | Graduate Training Grant 5T32GM007232-42 | Elizabeth A Turcotte |
| Brit d'Arbeloff MGH Research Scholar | | Cammie F Lesser |
| Medical Research Foundation | MRF2012 | Isabella Rauch |
| National Institutes of Health | AI063302 | Russell E Vance |
| National Institutes of Health | AI128743 | Cammie F Lesser |

The funders had no role in study design, data collection and interpretation, or the decision to submit the work for publication.

### Author contributions

Patrick S Mitchell, Justin L Roncaioli, Conceptualization, Formal analysis, Validation, Investigation, Methodology, Writing - original draft, Writing - review and editing; Elizabeth A Turcotte, Investigation, Writing - review and editing; Lisa Goers, Resources, Methodology, Writing - review and editing; Roberto A Chavez, Investigation; Angus Y Lee, Resources; Cammie F Lesser, Resources, Supervision, Funding acquisition, Methodology, Writing - review and editing; Isabella Rauch, Conceptualization, Resources, Supervision, Methodology, Writing - review and editing; Russell E Vance, Conceptualization, Resources, Supervision, Funding acquisition, Methodology, Writing - original draft, Writing - review and editing

### Author ORCIDs

Russell E Vance (iD) https://orcid.org/0000-0002-6686-3912

## Ethics

Animal experimentation: This study was performed in strict accordance with the recommendations in the Guide for the Care and Use of Laboratory Animals of the National Institutes of Health. All of the animals were handled according to approved institutional animal care and use committee (IACUC) protocols (AUP-2014-09-6665-1) of the University of California Berkeley.

## Decision letter and Author response

Decision letter https://doi.org/10.7554/eLife.59022.sa1
Author response https://doi.org/10.7554/eLife.59022.sa2

## Additional files

### Supplementary files
• Transparent reporting form

### Data availability

All data generated or analyzed during this study are included in the manuscript and supporting files.

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
