## [Decision Letter]

**Acceptance summary:**

In this manuscript, the authors introduce a new mouse model of Shigellosis, provide evidence for NAIP–NLRC4 activation as being important for epithelial cell defense, and apply these findings to observations made in humans infected by this pathogen. These are important findings and provide an opportunity to further advance the field in ways not previously possible. This is an important piece of work that will greatly influence the future directions of *Shigella* research. The revised manuscript also nicely addresses certain caveats in the model, while at the same time explaining how this model will advance the field.

**Decision letter after peer review:**

Thank you for submitting your article "NAIP-NLRC4-deficient mice are susceptible to shigellosis" for consideration by *eLife*. Your article has been reviewed by three peer reviewers, and the evaluation has been overseen by a Reviewing Editor and Gisela Storz as the Senior Editor. The following individual involved in review of your submission has agreed to reveal their identity: Edward A Miao (Reviewer #2).

The reviewers have discussed the reviews with one another and the Reviewing Editor has drafted this decision to help you prepare a revised submission.

We would like to draw your attention to changes in our revision policy that we have made in response to COVID-19 (https://elifesciences.org/articles/57162). Specifically, we are asking editors to accept without delay manuscripts, like yours, that they judge can stand as *eLife* papers without additional data, even if they feel that they would make the manuscript stronger. Thus the revisions requested below only address clarity and presentation. However, all three reviewers agree that if histology sections or serum samples were available, it would be nice to more closely examine pathology and cytokine levels in the model.

Summary:

In this manuscript, the authors introduce a new mouse model of Shigellosis, provide evidence for NAIP–NLRC4 activation as being important for epithelial cell defense, and apply these findings to observations made in humans infected by this pathogen. These are important findings and provide an opportunity to further advance the field in ways not previously possible. However, there are areas where the in vitro and in vivo data presented contradict each other, and there are inconsistencies with previously published work by the authors. In addition, with the development of the new mouse model being a major highlight of this manuscript, significantly more detail and discussion must be added to explain this mouse model. Therefore, the reviewers are requesting a major revision to the text of the manuscript to address these concerns, as detailed below.

Essential revisions:

1) Since the authors are presenting the paper largely as the first published mouse model for shigellosis, it is important to discuss the model's caveats for the benefit of others who may wish to utilize this model, as well as emphasize how much more there is to learn. A more in depth discussion about the differences between human *Shigella* infection and the new model would be helpful. It is important to emphasize that the mouse model requires a much greater inoculum of the pathogen to induce disease and requires microbiota-deficiency to be effective. What are the implications of this finding on our understanding of human disease? The authors present the inhibition of human NAIP–NLRC4 as the main factor that effects the difference in infection between humans and mice but a high innolcum (5 x 10(7) cfu/mouse compared to approx. 100 cfu for humans) is still required in addition to streptomycin treatment. It is not discussed whether any refinement of these procedures was attempted or why such a high inoculum and streptomycin treatment is still required. Presumably microbiota differences in addition to *Naip*-/*Nlrc4* is an important species specific determinant of infection, hence the streptomycin treatment.

2) In the second paragraph of the Discussion the authors present an either/or scenario in which either macrophage pyroptosis is required for IEC infection or inhibition of NAIP–NRLC4 pyroptosis in IECs is required for IEC infection. However, these scenarios are not mutually exclusive. For example, it is plausible that the extremely low burdens of *Shigella* required to infect humans (<100 CFUs) is due to the pathogen initially crossing the epithelial barrier (e.g. through M-cells) to infect macrophage, and then re-infection of IECs after macrophage pyroptosis. In this scenario, the NAIP–NLRC4 inflammasome could prevent further expansion of bacterial in IECs by eliminating the cell-to-cell spread that have been described by others. Importantly, the macrophage lifecycle stage may not be necessary in mice in which the microbiota has been removed and *Shigella* is delivered at a very high inoculum. While, additional ideas could be, and should be, put forth since the mouse model provides new insights or challenges an existing dogma in the field.

3) Many questions remain concerning why NLRC4-deficient THP1 cells still undergo pyroptosis. The authors provide evidence that *Shigella* activates PYRIN and/or AIM2 inflammasomes in humans, and that somehow mouse macrophages would fail to have this same detection. At face value, the data would suggest that humans are able to detect *Shigella* by PYRIN and AIM2, but for some reason these two inflammasomes are insufficient, and instead NLRC4 is required for in vivo defense. Then in mice, it would imply that everything is flipped – for some reason detection by PYRIN and AIM2 is not important, but now the bacteria can be detected by NLRC4 and this is important. The NLRC4 focused conclusions are consistent with the in vivo data, that NLRC4 in humans fails to detect, but NLRC4 in mice succeeds in detecting *Shigella*. However, the data that PYRIN and AIM2 in human cells successfully detect *Shigella* are inconsistent with the overall conclusions of the paper. I suspect that this is an artifact of THP1 cells, and that the in vivo situation in humans is that these two inflammasomes will fail to detect *Shigella*. There is published precedent from other infections where in vitro detection belies in vivo lack of detection (e.g. *Listeria* is detected by AIM2 in vitro, but probably not *in vivo*). It may be difficult to make direct comparisons between how inflammasomes act in THP1 cells as compared to BMMs, due to artifacts arising from the different origins and passage levels of the two cell types. It may be that the inflammasomes response is most important in IECs, as proposed by the authors, and that IECs may not express PYRIN or AIM2. There is evidence from publicly available IEC transcriptional profiles that IECs do not express PYRIN (*Mefv*) (Reikvam, doi: 10.1371/journal.pone.0017996), although this profile does show AIM2 expression in IEC. As it stands, the in vitro data appear to contradict one of the main conclusions of the paper, because it would seem that human PYRIN and AIM2 inflammasomes can detect *Shigella*, and so these should compensate for NLRC4. The explanation as to why PYRIN and AIM2 are insufficient to compensate for NLRC4 evasion in human infection should be addressed at least in discussions of the data to explain the apparent discrepancy.

4) The overall message of this manuscript directly contradicts the authors prior paper in Science just last year. They found that the *Shigella* effector IpaH7.8 directly and specifically ubiquitinates the mouse NLRP1B inflammasome, leading to activation of that inflammasome. How can the loss of NLRC4 be so important when mice still retain NLRP1B which the authors previously published to detect *Shigella*? Is this due to differences between murine NLRP1 alleles? This should be discussed in the text.

5) In the subsection “*Shigella* causes bloody diarrheal disease in 129.*Nlrc4*^–/–^ mice", the authors say that no blood was seen in B6 background, but they do not present direct data or discuss whether this statement arises from macroscopic examination (which would be unable to detect low level bleeding). Considering that a large proportion of the 129 mice only displayed occult bleeding, could the B6 mice have occult bleeding? This should at least be discussed clearly.

6) The authors have an entire section titled "Epithelial NLRC4 protects mice from shigellosis via IEC expulsion," but they have only demonstrated that epithelial NLRC4 is sufficient to protect from shigellosis. As the Vance group itself has previously demonstrated (Rauch et al., 2017), inflammasome activation in intestinal epithelial cells has multiple effects, including pyroptosis, epithelial expulsion, and inflammatory cytokine release, all of which may (or may not) contribute to defense. Thus, their data cannot actually attribute the role of NLRC4 to epithelial expulsion. The title of the section needs to be modified.

7) If available, the following data would increase the impact of the manuscript:

A) Demonstration that IECs from NLRC4^-/-^ mice undergo cell death.

B) Bacterial burden over the time course in Figure 6.

C) Measurement of markers of inflammation and histopathology during disease. Since Shigellosis is an inflammatory disease, it would have been nice to have seen some inflammatory molecules/cytokine levels measured, in addition to clinical features. The authors did measure levels of MPO, but that was as a marker for neutrophil recruitment. It would also be great to have a more thorough comparison of the histology observed in this mouse model as compared to human shigellosis. This could be a Discussion point as something that needs to be further considered in future work, but it should be pointed out to the reader who may be considering adopting this model that this step has not yet been taken.

---

## [Author Response]

Essential revisions:1) Since the authors are presenting the paper largely as the first published mouse model for shigellosis, it is important to discuss the model's caveats for the benefit of others who may wish to utilize this model, as well as emphasize how much more there is to learn. A more in depth discussion about the differences between human Shigella infection and the new model would be helpful. It is important to emphasize that the mouse model requires a much greater inoculum of the pathogen to induce disease and requires microbiota-deficiency to be effective. What are the implications of this finding on our understanding of human disease? The authors present the inhibition of human NAIP–NLRC4 as the main factor that effects the difference in infection between humans and mice but a high innolcum (5 x 10(7) cfu/mouse compared to approx. 100 cfu for humans) is still required in addition to streptomycin treatment. It is not discussed whether any refinement of these procedures was attempted or why such a high inoculum and streptomycin treatment is still required. Presumably microbiota differences in addition to naip-/nlrc4 is an important species specific determinant of infection, hence the streptomycin treatment.

We agree with the reviewer that the infectious dose in humans vs. mice merits more discussion in a revised manuscript. We used a dose of 5x10^7^ CFU in our manuscript to ensure reproducible infection of all mice. Although it is often (correctly) stated that as few as 10-100 bacteria *can* infect humans with *Shigella*, there is actually considerable heterogeneity in the infectious dose. DuPont et al., 1989, summarizes several human challenge studies in their Table 1, which shows that while 25-39% of humans exhibit symptoms after low dose infection (<200 CFU), 36-44% of humans are resistant to high doses (10^4^-10^8^ CFU). Thus we do not consider the infectious dose in our mouse model to be out of the range of what is ‘normal’ in humans. Indeed, our new model may help us understand some of the factors that confer resistance to certain humans.

That being said, we have now conducted experiments to address the necessary infectious dose in our model, using inoculums of 2x10^4^, 2x10^5^, and 2x10^6^ in NAIP–NLRC4-deficient mice compared to 2x10^6^ in WT mice. We observed fully penetrant disease in NAIP–NLRC4-deficient mice inoculated with as few as 2x10^4^ CFU. These data are included in a new Figure 8. We also observed that *Shigella* markedly expands in the antibiotic-cleared gut lumen, regardless of the starting inoculum, as early as 1d post-infection (Figure 8A), suggesting that there is a threshold dose above which the outcomes are largely similar.

Antibiotic pre-treatment is a common feature of many mouse models for enteric pathogens, including a very widely used model of *Salmonella* infection, but we agree this is important to highlight and have now added discussion to the manuscript to call attention to this point. We hope that this satisfies the concern of the reviewers regarding inoculum.

2) In the second paragraph of the Discussion the authors present an either/or scenario in which either macrophage pyroptosis is required for IEC infection or inhibition of NAIP–NRLC4 pyroptosis in IECs is required for IEC infection. However, these scenarios are not mutually exclusive. For example, it is plausible that the extremely low burdens of Shigella required to infect humans (<100 CFUs) is due to the pathogen initially crossing the epithelial barrier (e.g. through M-cells) to infect macrophage, and then re-infection of IECs after macrophage pyroptosis. In this scenario, the NAIP–NLRC4 inflammasome could prevent further expansion of bacterial in IECs by eliminating the cell-to-cell spread that have been described by others. Importantly, the macrophage lifecycle stage may not be necessary in mice in which the microbiota has been removed and Shigella is delivered at a very high inoculum. While, additional ideas could be, and should be, put forth since the mouse model provides new insights or challenges an existing dogma in the field.

We do clearly state in our manuscript (Discussion, fourth paragraph) that our results do not directly address the question of whether *Shigella* might benefit from inflammasome activation in macrophages. In the revised version of the manuscript we have further expanded on the discussion of the role of inflammasomes in macrophages and IECs to acknowledge multiple, non-mutually exclusive scenarios. We have also now measured cytokines from tissue and find that IL-1β is elevated in *Nlrc4^–/–^* mice (Figure 3F). We now also discuss the possibility that, in the absence of NAIP–NLRC4, the elevated IL-1β reflects inflammasome activation in *Shigella*-infected macrophages on the basolateral side of the gut epithelium.

3) Many questions remain concerning why NLRC4-deficient THP1 cells still undergo pyroptosis. The authors provide evidence that Shigella activates PYRIN and/or AIM2 inflammasomes in humans, and that somehow mouse macrophages would fail to have this same detection. At face value, the data would suggest that humans are able to detect Shigella by PYRIN and AIM2, but for some reason these two inflammasomes are insufficient, and instead NLRC4 is required for in vivo defense. Then in mice, it would imply that everything is flipped – for some reason detection by PYRIN and AIM2 is not important, but now the bacteria can be detected by NLRC4 and this is important. The NLRC4 focused conclusions are consistent with the in vivo data, that NLRC4 in humans fails to detect, but NLRC4 in mice succeeds in detecting Shigella. However, the data that PYRIN and AIM2 in human cells successfully detect Shigella are inconsistent with the overall conclusions of the paper. I suspect that this is an artifact of THP1 cells, and that the in vivo situation in humans is that these two inflammasomes will fail to detect Shigella. There is published precedent from other infections where in vitro detection belies in vivo lack of detection (e.g. Listeria is detected by AIM2 in vitro, but probably not in vivo). It may be difficult to make direct comparisons between how inflammasomes act in THP1 cells as compared to BMMs, due to artifacts arising from the different origins and passage levels of the two cell types. It may be that the inflammasomes response is most important in IECs, as proposed by the authors, and that IECs may not express PYRIN or AIM2. There is evidence from publicly available IEC transcriptional profiles that IECs do not express PYRIN (Mefv) (Reikvam, doi: 10.1371/journal.pone.0017996), although this profile does show AIM2 expression in IEC. As it stands, the in vitro data appear to contradict one of the main conclusions of the paper, because it would seem that human PYRIN and AIM2 inflammasomes can detect Shigella, and so these should compensate for NLRC4. The explanation as to why PYRIN and AIM2 are insufficient to compensate for NLRC4 evasion in human infection should be addressed at least in discussions of the data to explain the apparent discrepancy.

The reviewer states that our data that human PYRIN and AIM2 inflammasomes can detect *Shigella* in THP1 cells are “inconsistent” or “appear to contradict” the overall conclusion of our paper, which is that the NLRC4 inflammasome provides necessary defense of mouse intestinal epithelial cells. For reasons explained below, we do not agree that there is an inconsistency or contradiction and indeed many of the points the reviewer makes in their comments fit with our view, so perhaps there is less disagreement than it might seem. As the reviewer discusses, differences in inflammasome expression in humans vs. mice, and in IECs vs. macrophages vs. THP1 cells, and the kinetics of inflammasome responses, as well as several other factors, can easily account for the results we obtain. It appears that PYRIN is not well expressed in mouse IECs (Price et al., 2016), at least not uniformly at levels in all cells that are sufficient to confer protection. AIM2 is expressed in colonic IECs (Price et al., 2016), but it is not clear that it would be engaged in every infected IEC. For example, AIM2 detects bacterial DNA, which might only be released if the *Shigella* bacteria lysed in the cytosol. As noted by the reviewer, this may be a relatively rare event, as previously documented for AIM2 activation by *Listeria*-infected macrophages (Sauer JD et al., 2010). AIM2 activation may also be kinetically delayed in IECs.

It appears instead that NLRC4 is the main inflammasome that can respond to *Shigella* in mouse IECs; thus loss of NLRC4 is sufficient to lead to susceptibility of mice. It remains possible that there is some functional AIM2 or PYRIN (or CASP11 or NLRP1B) in mouse IECs; thus, the further removal of these inflammasomes might lead to even greater susceptibility. Alternatively, a low level of activation mediated by these additional inflammasomes (perhaps in macrophages instead of in IECs) might even be necessary to produce the inflammation that causes disease symptoms.

In humans, consistent with our data in Figure 1, we propose that the NLRC4 inflammasome is antagonized or otherwise evaded by *Shigella*. The reviewer wonders why PYRIN or AIM2 cannot compensate for NLRC4, and is suspicious that the activation of PYRIN/AIM2 we observe in THP1 cells is not representative of what would occur in vivo. Certainly we agree that THP1 cells are non-physiological and we do not attempt to make claims in the manuscript that our observation of AIM2/PYRIN activity in these cells means anything for human shigellosis.

The reviewer states: “the in vitro data [in THP1 cells] appear to contradict one of the main conclusions of the paper, because it would seem that human Pyrin and AIM2 inflammasomes can detect *Shigella*, and so these should compensate for NLRC4.” For all the reasons discussed above, we do not agree there is a contradiction. There are many reasons why PYRIN and AIM2 might function in THP1 cells (and possibly even human macrophages) but would not compensate for NLRC4 in IECs.

In sum, we agree that there is more to learn about which inflammasomes, if any, are activated by *Shigella* in human IECs, but given the many uncertainties, we do not feel it is fair to say that our results are internally contradictory. We now discuss some of these points in a revised manuscript.

We would also like to share new data with the reviewers (see Author response image 1). We have now made *MEFV/AIM2^–/–^* THP-1 cells (*MEFV* is the gene encoding PYRIN). We find that *Shigella*-induced cell death and IL-1β levels in *MEFV/AIM^–/–^* is greatly reduced, suggesting that MEFV and AIM2 are responsible for the cell death in *Shigella* infected cells. This also supports the idea that the NAIP–NLRC4 inflammasome is inactive during *Shigella* infection. Moreover, repeating the suppression assay in *MEFV/AIM^–/–^* clearly demonstrates that NeedleTox fails to activate the NAIP–NLRC4 inflammasome following infection with WT but not BS103 *Shigella*. However, because we were only able to generate a single DKO clonal line, we are being cautious in our interpretation and would not like to publish these data at the present time. We also concede that the discussion of Figure 1E, and the use of chemical inhibitors instead of genetic knockouts, limits our ability to make firm conclusions. We thus word our discussion of these data very cautiously and directly state that further follow up work will be required. The main purpose in our view is simply to set up the hypothesis that if there is a species-specific function of NLRC4 in restriction of *Shigella*, then loss of NLRC4 should render mice susceptible to *Shigella* – a hypothesis that the rest of the paper then addresses.

4) The overall message of this manuscript directly contradicts the authors prior paper in *Science* just last year. They found that the *Shigella* effector IpaH7.8 directly and specifically ubiquitinates the mouse NLRP1B inflammasome, leading to activation of that inflammasome. How can the loss of NLRC4 be so important when mice still retain NLRP1B which the authors previously published to detect Shigella? Is this due to differences between murine NLRP1 alleles? This should be discussed in the text.

We respectfully disagree with the assertion that our current findings contradict our prior paper in *Science*. IpaH7.8 exclusively activates the 129 variant of mouse NLRP1B. Thus, we would not expect NLRP1B to play a role on the B6 background. We also note that it is currently unknown whether or not *Nlrp1b* is functional in IECs in 129 or B6 mice. Interestingly, CASP11, which is known to be both expressed and functional in IECs, does not compensate for a loss of NLRC4. Thus, the presence of a functional inflammasome (whether it be NLRP1B or CASP11) may not necessarily compensate for the function of NAIP–NLRC4 in IECs. As has been pointed out by the reviewers, it may also be possible that *Shigella* utilizes inflammasome activation in macrophages, a possibility that our current manuscript does not address. Thus, NLRP1B in macrophages may respond to *Shigella* and promote the more pronounced phenotype we observe on the 129 background. These are follow up questions that we are working on but we do not feel they are directly relevant to establish the claims of the current manuscript.

5) In the subsection “Shigella causes bloody diarrheal disease in 129.Nlrc4^–/–^ mice", the authors say that no blood was seen in B6 background, but they do not present direct data or discuss whether this statement arises from macroscopic examination (which would be unable to detect low level bleeding). Considering that a large proportion of the 129 mice only displayed occult bleeding, could the B6 mice have occult bleeding? This should at least be discussed clearly.

Thank you for this suggestion. We now discuss in the text that we are unable to detect occult blood in the feces of B6 NAIP–NLRC4-deficient mice.

6) The authors have an entire section titled "Epithelial NLRC4 protects mice from shigellosis via IEC expulsion," but they have only demonstrated that epithelial NLRC4 is sufficient to protect from shigellosis. As the Vance group itself has previously demonstrated (Rauch et al., 2017), inflammasome activation in intestinal epithelial cells has multiple effects, including pyroptosis, epithelial expulsion, and inflammatory cytokine release, all of which may (or may not) contribute to defense. Thus, their data cannot actually attribute the role of NLRC4 to epithelial expulsion. The title of the section needs to be modified.

We have modified the title of this section to “Epithelial NLRC4 protects mice from shigellosis.” However, we have also made a very substantive addition to the manuscript in which we demonstrate, using organoids, that IECs are indeed expulsed from the epithelium in response to *Shigella* in an NLRC4-dependent manner. These new data are now shown in Figure 6.

7) If available, the following data would increase the impact of the manuscript:A) Demonstration that IECs from NLRC4^-/-^ mice undergo cell death.

We presume the reviewer means to ask us to show that IECs undergo NLRC4-dependent cell death (which is what we hypothesize must occur; whether or not IECs die in the absence of NLRC4 remains an open question. We suspect that this does occur eventually as a result of bacterial replication-induced lysis). To address the reviewers request, and consistent with our overall model, we now provide new data showing that PI-positive (dead) cells from *Shigella*-infected monolayers are expelled from IEC monolayer cultures in an NLRC4dependent manner. We now provide these data in Figure 6C-E.

B) Bacterial burden over the time course in Figure 6.

We appreciate that the inclusion of CFU data across multiple time points post-infection would add some information to our manuscript. However, given current restrictions and the considerable down-sizing of our mouse colony during the shelter-in-place order, we simply do not have the requisite numbers of mice to accommodate this request. Regardless, we do not feel such data are necessary to establish the main claims of our manuscript.

C) Measurement of markers of inflammation and histopathology during disease. Since Shigellosis is an inflammatory disease, it would have been nice to have seen some inflammatory molecules/cytokine levels measured, in addition to clinical features. The authors did measure levels of MPO, but that was as a marker for neutrophil recruitment. It would also be great to have a more thorough comparison of the histology observed in this mouse model as compared to human shigellosis. This could be a Discussion point as something that needs to be further considered in future work, but it should be pointed out to the reader who may be considering adopting this model that this step has not yet been taken.

We agree that additional readouts of inflammatory disease are warranted. We now present levels of pro-inflammatory cytokines IL-1β, IL-18 and KC from intestinal tissue in Figure 3H.